# Deep imaging reveals dynamics and signaling in one-to-one pollen tube guidance

Yoko Mizuta [1,2✉], Daigo Sakakibara[3], Shiori Nagahara[2], Ikuma Kaneshiro[2,4,5], Takuya T Nagae[3], Daisuke Kurihara [1,2] & Tetsuya Higashiyama[2,3,6]

## Abstract

In the pistil of flowering plants, each ovule usually associates with a single pollen tube for fertilization. This one-to-one pollen tube guidance, which contributes to polyspermy blocking and efficient seed production, is largely different from animal chemotaxis of many sperms to one egg. However, the functional mechanisms underlying the directional cues and polytubey blocks in the depths of the pistil remain unknown. Here, we develop a two-photon live imaging method to directly observe pollen tube guidance in the pistil of *Arabidopsis thaliana*, clarifying signaling and cellular behaviors in the one-to-one guidance. Ovules are suggested to emit multiple signals for pollen tubes, including an integument-dependent directional signal that reaches the inner surface of the septum and adhesion signals for emerged pollen tubes on the septum. Not only FERONIA in the septum but ovular gametophytic FERONIA and LORELEI, as well as FERONIA- and LORELEI-independent repulsion signal, are involved in polytubey blocks on the ovular funiculus. However, these funicular blocks are not strictly maintained in the first 45 min, explaining previous reports of polyspermy in flowering plants.

**Keywords** Sexual Reproduction; Pollen Tube; Polyspermy Block; Two-Photon Microscopy; *Arabidopsis thaliana*
**Subject Categories** Development; Plant Biology

## Introduction

Flowers are the reproductive organs of angiosperms that evolved between the Jurassic and Lower Cretaceous Periods (Li et al, 2019). Within a flower, two sperm cells fuse with an egg and a central cell, initiating seed development. However, sperm cells have lost their motility to evolution and require the pollen tube to be delivered to the ovule, which is deeply embedded in the pistil base (Dresselhaus et al,

2016). To ensure successful fertilization, angiosperms have evolved multiple mechanisms for pollen–pistil interactions involving ovular chemotropism (Mizuta and Higashiyama, 2018). Chemotropism is the directed growth of cells, tissues, or organisms in response to external cues, such as the axon guidance of olfactory sensory neurons expressing individual odorant receptors (Imai et al, 2006; Reinert and Fukunaga, 2022). Angiosperms employ a chemotropic guidance system known as "pollen tube guidance." In the flower of *Arabidopsis thaliana*, a gynoecium consists of two fused carpels. Each carpel represents one locule of the ovary that contains dozens of linearly aligned ovules (Fig. EV1A). When pollen lands on the stigma, the pollen hydrates, leading to the germination of the pollen tube, which then penetrates the ovary. After the pollen tube enters the transmitting tract (TT) within the ovary, it reaches the ovule via three steps: emergence from the TT, growth on the funiculus (funicular guidance), and penetration of the micropyle (micropylar guidance) (Fig. EV1A, right). During this process, the fertilization system termed "polytubey block" functions to prevent multiple pollen tubes from penetrating a single ovule (Sugi and Maruyama, 2023). This restricted fertilization of the egg by more than one sperm (polyspermy) maximizes reproductive success and produces more offspring (Spielman and Scott, 2008). Genetic studies have demonstrated that ovules secrete small defensin-like peptides (LUREs and LURE-like peptides) that act as chemical attractants for pollen tubes (Meng et al, 2019; Takeuchi and Higashiyama, 2012; Takeuchi and Higashiyama, 2016; Zhong et al, 2019). As such, ovule-derived chemoattractants can affect all pollen tubes, it is thought that an ovule can potentially attract multiple pollen tubes. However, polytubey is rare in *A. thaliana* (~1%) (Huck et al, 2003; Rotman et al, 2003), despite the presence of ~60 ovules and >60 pollen tubes (Yuan and Kessler, 2019).

Recently, some genetic approaches have demonstrated that multi-step polytubey blocking systems regulate one-to-one pollen tube guidance. FERONIA (FER) is a receptor-like kinase that regulates some important plant signaling pathways, including development, immune/stress response, and pollen tube reception (Ji et al, 2020). FER-dependent blocks affect pollen tube emergence as a sporophytic signal in the septum (Zhong et al, 2022), and suppress polytubey as a gametophytic signal in the synergid cells (Duan et al, 2020). Once a pollen tube reaches the micropyle, the further attraction of pollen tubes

[1]Institute for Advanced Research (IAR), Nagoya University, Furo-cho, Chikusa-ku, Nagoya, Aichi 464-8601, Japan. [2]Institute of Transformative Bio-Molecules (WPI-ITbM), Nagoya University, Furo-cho, Chikusa-ku, Nagoya, Aichi 464-8601, Japan. [3]Division of Biological Sciences, Graduate School of Science, Nagoya University, Furo-cho, Chikusa-ku, Nagoya, Aichi 464-8602, Japan. [4]Department of Physics, Graduate School of Science, Nagoya University, Furo-cho, Chikusa-ku, Nagoya, Aichi 464-8602, Japan. [5]Research Center for Computational Science, Institute for Molecular Science, National Institutes of Natural Sciences, 38 Nishigo-Naka, Myodaiji, Okazaki 444-8585, Japan. [6]Present address: Department of Biological Sciences, Graduate School of Science, The University of Tokyo, 7-3-1 Hongo, Bukyo-ku, Tokyo 113-0033, Japan.
✉E-mail: mizuta.yoko.u6@f.mail.nagoya-u.ac.jp

is ceased due to the absence or inactivation of pollen tube attractants, including LUREs (Duan et al, 2020; Maruyama et al, 2015; Völz et al, 2013; Yu et al, 2021). However, because the real-time behavior of pollen tube attraction and the temporal relationships of these multiple blocks have not been observed, the full context of the blocking system remains unclear.

In the present study, we developed an imaging method using two-photon excitation microscopy based on a previous method (Rotman et al, 2003), aiming to investigate pollen tube dynamics in living ovaries of *A. thaliana*. Genetic approaches with imaging analysis reveal multilayered regulation by the ovule to achieve one-to-one pollen tube guidance. Our study provides new insights into how plants maximize offspring production through successful fertilization.

# Results

## Single-locule method enables two-photon live imaging of one-to-one pollen tube guidance in a living ovary

Under our experimental conditions, the number of ovules in a flower of the wild-type (WT) Col-0 ecotype of *A. thaliana* was 46.8 ± 3.3 (mean ± s.d.; *n* = 10 pistils). The number of pollen tubes in a maximum pollinated pistil was counted 18 h after pollination (HAP) when all three steps of guidance (emergence from the TT, growth on the funiculus, and penetration of the micropyle) had been completed in almost all ovules. NaOH-cleared pollinated pistils stained with aniline blue solution was used to count the number of pollen tubes growing in the TT, which was 70 ± 3.6 (mean ± s.d.; *n* = 3 pistils; Fig. EV1B−F). Thus, the number of pollen tubes per ovule is approximately 1.5, indicating that almost all ovules must attract a pollen tube on a one-to-one basis for all ovules to be fertilized. In the NaOH-cleared pistil, the pollen tubes within the TT at maximum pollination looked like a pollen tube "bundle" (Fig. EV1B). Some pollen tubes were shown to grow in the *z*-direction from such a pollen tube bundle, which was thought to be an attracted pollen tubes to the ovule (Fig. EV1C). However, the process of pollen tube guidance on each pollen tube, especially the temporal sequence, was difficult to analyze in the fixed tissue (Fig. EV1B). Thus, we developed a two-photon live imaging method named, "single-locule method" for deep imaging of pollen tube dynamics within a live *Arabidopsis* ovary expressing fluorescent reporter (Fig. EV2). The pollen tube bundle that entered from style into the ovary TT reached the base of the WT ovary at 16 HAP in the WT ovary (Movie EV1A; Fig. 1A–C). The pollen tube growth rate was not constant and gradually decreased to 1.3 ± 0.1 μm/min (mean ± s.d.; *n* = 7 pollen tubes) between 4 and 16 HAP (Fig. EV3A). In the *yz*-projection image, some pollen tubes abruptly separated from pollen tube bundle to pass the septum epidermal (SE) layer, adjusting their growth towards the locule (Fig. 1C). Such directional changes were assumed to indicate pollen tube emergence followed by funicular guidance, as the fluorescent signals of these tubes showed that they elongated along the funiculus after passing through the septum epidermis (SE) (Fig. EV2C), being attracted to the fluorescent-labeled synergid cells (Fig. 1D–G; Movies EV1B,C). Time-lapse imaging of the area encompassing 512 μm from the upper region of the ovary of the limited or maximum pollinated pistils was performed using the single-locule method. In the 314 WT pistils for 1 to 18 h after pollination, pollen tube growth in the TT was observed in 218, and pollen tube

emergence and/or funicular guidance was observed in 110 of the pistils with 448 ovules. The two-photon live imaging separately analyzed the processes of pollen tube elongation, pollen tube emergence, and funicular guidance in a living ovary. To the best of our knowledge, this is the first report of such a dynamic recording of the entire guidance of a single pollen tube. Therefore, our single-locule method enabled real-time analysis of pollen tube guidance until it reached the micropyle.

## Pollen tube emergence is stochastically regulated by the number and distribution of pollen tubes

In *A. thaliana*, it is unclear which ovule would preferentially attract a pollen tube among ovules (Crawford et al, 2007; Hülskamp et al, 1995). As the ovules are linearly aligned and pollen tubes elongate from the top, we hypothesized that pollen tube guidance begins at the apical ovules. To confirm this hypothesis, NaOH-cleared pollinated pistils were observed (Fig. 2). Under maximum pollination, pollen tubes were distributed throughout the entire TT (Fig. EV1B–F). As expected, under maximum pollination, the median distribution of the targeted ovules was 22.0% (205 ovules in 12 pistils, Fig. 2D), which were distributed towards the apex. However, when single pollen pollination was performed (limited pollination, Fig. 2A,B), such apical preferentiality was lost. The single seed distribution in the silique indicated that the germinated single pollen tube had reached it (Fig. 2C). The apical 10% of the ovules were not targeted in both center and side pollination conditions (Fig. 2E,F). The median single seed distribution from the top of the silique was 49% for central pollination (*n* = 52 siliques, Fig. 2E) and 46% for side pollination (*n* = 44 siliques, Fig. 2F), both of which were more located at the bottom of the silique than the maximum pollination. No difference in targeting preferentiality was observed between the central and side pollinations, and the apical 10% of ovules were not targeted in either condition (Fig. 2E,F). Therefore, preferential ovule targeting depended on the number of germinated pollen grains, but not on the position of the pollen grain on the stigma. Moreover, the results suggest that a pollen tube can pass the apical ovules, implying that an ovular attraction signal may only be effective locally. In other words, pollen tubes may need to reach the zone of ovular attraction in the TT to be attracted to an ovule. Therefore, we assumed that growing pollen tubes emerge upon receiving some local signals, and next, accordingly analyzed the position of the pollen tube and the time of pollen tube emergence by two-photon live imaging.

Two-photon live imaging showed that pollen tubes are allowed to grow freely by changing available radial positions in the TT under limited pollination (Fig. 3A–D). Pollen tube behavior and growth path were investigated using an SE reporter (Ueda et al, 2017) to determine which pollen tube emerged from the TT (Fig. EV2B,C). Under limited pollination, some pollen tubes gradually moved toward the SE and emerged from the TT, whereas other pollen tubes grew down the TT (Fig. 3A,B). Such gradual movement of the pollen tubes toward the SE and then elongates along the SE within the TT, was observed in many pistils examined, even in the presence of multiple pollen tubes in the TT (white arrow in Fig. 3D, cyan, green, and orange arrows in Fig. 4B; see also Movie EV2). When two pollen tubes elongated in parallel, the one closer to the SE was more likely to emerge (Fig. 3C,D). Therefore, we concluded that pollen tubes were attracted from the top down under conditions of maximum pollination but not limited pollination, for two reasons: pollen tubes adjacent to the SE were selected for emergence in the TT, and under maximum pollination,

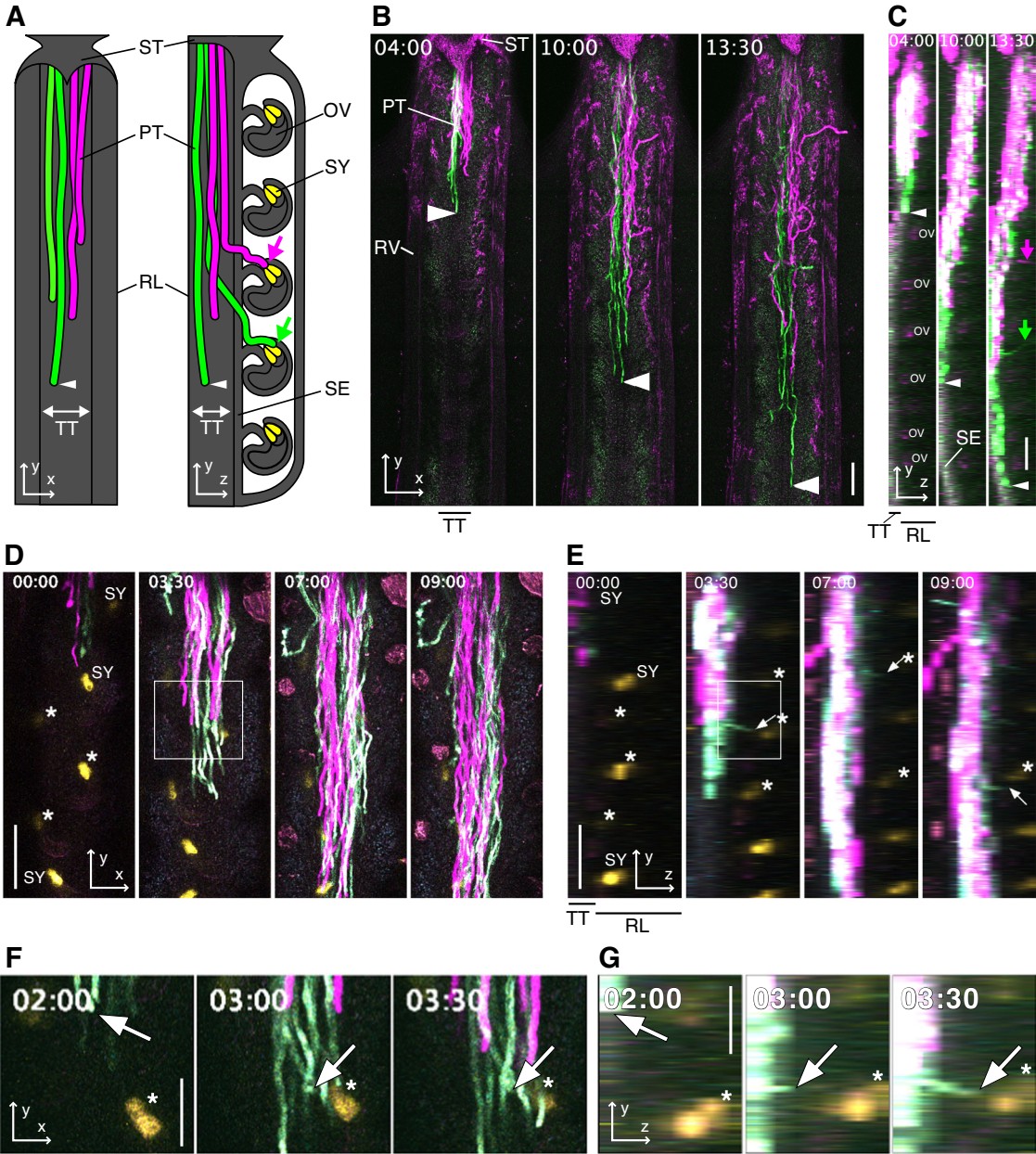

**Figure 1. Pollen tube dynamics in the *Arabidopsis* living pistil.**

(A) Schematic representations of the *xy*- (left) and *yz*- (right) images of pollen tube growth in the ovary under the single-locule method. (B, C) Wild-type pistils pollinated with a mixture of pollen expressing mTFP1 (green) and TagRFP (magenta). The *xy*- (B) and *yz*- (C) maximum projection images by 10-µm steps including transmitting tract (TT) are shown. Magenta and green arrows indicate pollen tubes guided toward the ovule after emergence from the TT. (D, E) The pistil having ovules with synergid cells labeled by GFP (yellow) was pollinated with the pollen expressing mTFP1 (cyan) and TagRFP (magenta). The *xy*- (D) and *yz*- (E) maximum projection images by 7 *z*-stack images with 15-µm steps are shown. (F, G) Magnified images of the funicular guidance are highlighted by a white box in (D) and (E). Arrowheads indicate the tip of most growing pollen tubes. Arrows indicate pollen tubes guided toward the ovule after emergence from the TT. Asterisks indicate synergid cells of the target ovules. Data Information: The numbers stamped in each frame indicate the time (h:mm) from the start of the observation. ST style, PT pollen tube, RL remaining locule, TT transmitting tract, OV ovule, SY synergid cell, SE septum epidermis. Scale bars, 100 µm (B–E) and 50 µm (F, G). See also Movie EV1. Source data are available online for this figure.

pollen tubes were more likely to grow adjacent to the SE. These results suggest that pollen tube emergence is stochastically determined depending on the growth position in the TT; the closer pollen tubes grow to the SE, the more pollen tubes are attracted and emerge by sensing some local signals adjacent to the SE.

## Ovule maturation can affect pollen tubes within the TT and contribute to ovule targeting

To assess the properties of the emergence signal(s) adjacent to the SE, we investigated the temporal regulation of pollen tubes that had emerged. As mentioned above, two-photon live imaging of the 110 WT pistils for 1 to

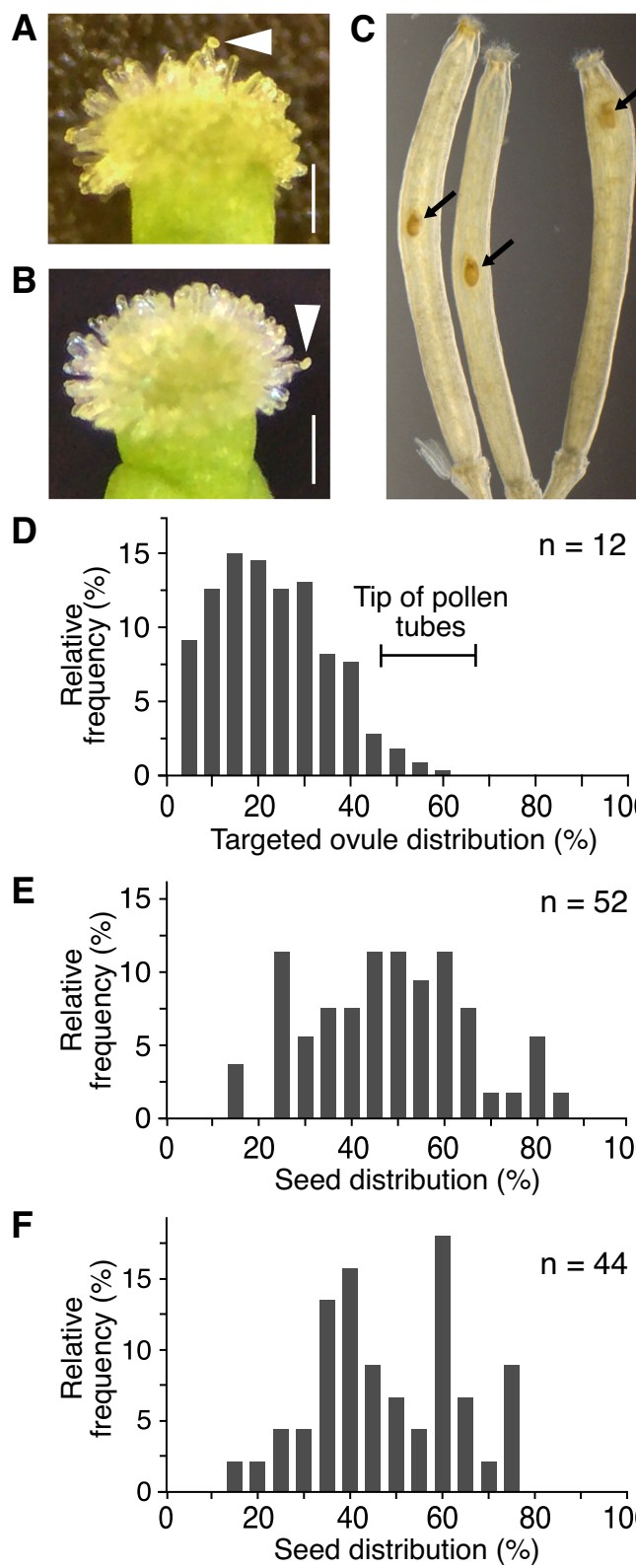

**Figure 2.** Emergence and ovule targeting of the pollen tube from the transmitting tract under limited pollination.

(A, B) Spatially regulated limited pollination. Pollinated single pollen grains are shown as arrowheads at the center (A) or side (B) of the stigma. (C) Transparent siliques with one seed from each pollination of single pollen. Seeds are indicated by arrows. (D) Relative frequency of the distributions of the 205 targeted ovules in the maximum pollinated 12 pistils, when the tip of the pollen tube reached the middle part at 4 h after pollination (56.5 ± 7.1% from the top of the ovary; $n = 12$ pistils). (E, F) Single seed-set distribution of the 52 pistils with center pollination (E) and 44 pistils with side pollination (F) under the single pollen pollination. Bar charts represent the frequency of seeds present in each of the ten percentiles of ovary length, from the most apical (0–10%) to the most basal (90–100%) of each ovary. Data information: In (D), data were presented as mean ± s.d. In (E) and (D), we used the non-parametric Mann–Whitney $U$-tests for the statistical analysis, and the difference between center and side pollinations was not significant ($P = 0.99$). Source data are available online for this figure.

decreased to 0.5 μm/min ~2 h before their emergence (Fig. 4A), while non-emerged tubes maintained a growth rate of 1.4 ± 0.3 μm/min (mean ± s.d.; $n = 69$ pollen tubes in ten pistils). The $xy$- and $yz$-projection images revealed that the pollen tube growth rate decreased following its attachment to the SE within the TT (Fig. 4B,C; EV3B). The median time from pollen tube SE attachment until emergence was 120 min ($n = 11$; range 45–285 min), consistent with the decreased growth rate. The growth rates of the emerged nine pollen tubes on both septum and funiculus after pollen tube emergence were measured from $yz$-projection images. The growth rates on the septum and funiculus were 0.5 ± 0.06 μm/min and 0.6 ± 1.6 μm/min, respectively (mean ± s.d.; $n = 9$ pollen tubes). This result suggests that SE attachment within the TT triggers a reduction in the pollen tube growth rate, persisting throughout its emergence until growing on the septum. However, subsequent funicular guidance showed variation in pollen tube growth rates on the funiculus (0.6 ± 1.6 μm/min). These results suggest that different factors affect pollen tube growth on the septum and on the funiculus.

Ovular cells, especially the two synergid cells, are indispensable for ovule targeting (Higashiyama and Takeuchi, 2015; Lopes et al, 2019). Moreover, soluble secreted materials from the mature ovule act as long-distance pollen tube guidance signals in *Torenia* (Horade et al, 2013). To investigate the effect of ovule maturity on the pollen tube emergence from the TT, ovule development was examined in ClearSee-treated cleared pistils (Kurihara et al, 2015; Mizuta and Tsuda, 2018). Ovule maturation was indicated with the appearance of a fluorescent signal of the FGR8.0 reporter, which combines ovular cell fluorescent markers (Völz et al, 2013). Maturation commenced in the middle to lower portion of the ovary at stage 11, then extended to both the upper and lower extremities until stage 14 (Fig. EV4A). Even on the day of flowering, at stage 14, some ovules in the topmost ovary remained immature (arrows in Fig. EV4A). This result correlated with the ovule targeting preference under limited pollination conditions (Figs. 2 and 3), suggesting that ovule maturation contributes to ovule targeting. It was also consistent with our hypothesis that the emergence signal(s) are derived from the ovules and reach to the inner surface of the TT.

## Long-distance guidance signal depending on ovular sporophytic outer integument regulates pollen tube emergence

The live imaging results suggest that pollen tubes in the TT are guided before they emerge, and that ovule maturation may be involved in

18 HAP revealed 448 ovules with pollen tube attraction. Of these, nine pollen tubes in seven pistils were able to be analyzed the growth rate of pollen tubes within the TT from more than 2.5 h before pollen tube emergence (Fig. 4A). The growth rate of emerged pollen tubes gradually

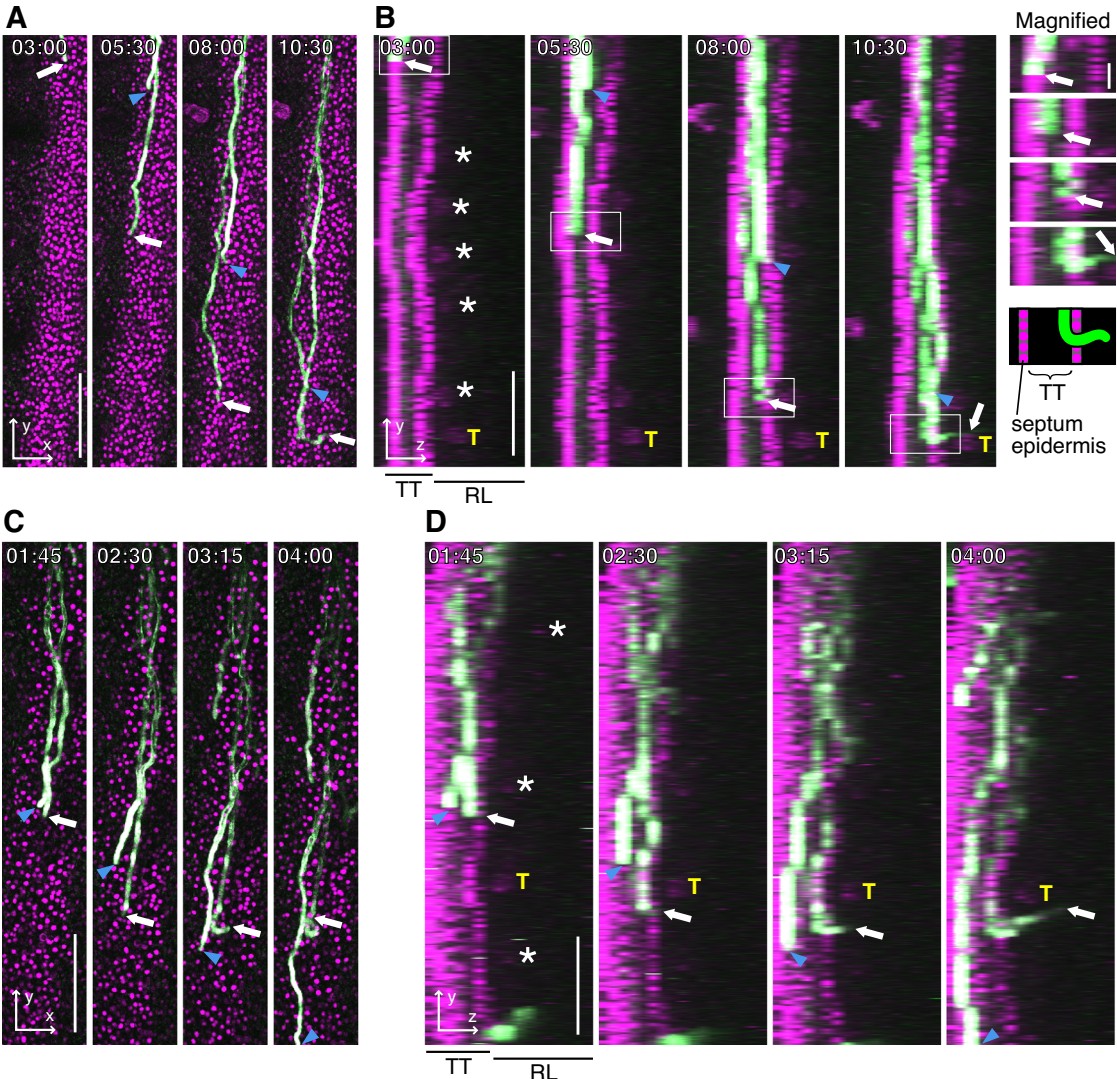

**Figure 3. Live imaging of pollen tube emergence and ovule targeting under limited pollination.**

(A–D) Emergence of the pollen tube from the transmitting tract. Pollen tubes and nuclei of the septum epidermis labeled with each fluorescent protein are shown in green and magenta, respectively. See also Movies EV2A and EV2B. The *xy*- (A, C) and *yz*- (B, D) projection images by 10-μm steps with 18 planes. Magnified images with a schematic representation of the emergence of pollen tube from the transmitting tract highlighted by a white box in (B). (C, D) The pollen tube at the closest to the septum epidermis emerged from the transmitting tract. Data Information: Arrows and arrowheads indicate emerged and non-emerged pollen tubes, respectively. Asterisks show the autofluorescence of the ovule. Time stamps are the same as in Fig. 1. T targeted ovule, TT transmitting tract, RL remaining locule. Scale bars, 100 μm, and 20 μm in the magnified image of (B). Source data are available online for this figure.

pollen tube emergence. However, long-range ovular guidance has been a subject of debate (Higashiyama and Takeuchi, 2015). The female gametophyte and its synergid cell are the source of the ovular attraction signal (Takeuchi, 2021). The central cell has also been reported to be involved in fertilization recovery when the first tube fails fertilization (Meng et al, 2023). Due to the observed correlation between ovule maturation and pollen tube targeting under limited pollination conditions as described above, we hypothesized that the female gametophyte might be the origin of the long-distance emergence signal that reaches the inner surface of the septum. Thus, ClearSee-treated cleared pistils of the WT and three ovular mutants were examined. The ovular tissues of each sample are as follows; (1) WT has both sporophytic and gametophytic tissues in an ovule, (2) In

the *aintegumenta* (*ant*) mutant ovule, both sporophyte and gametophyte are severely defective, due to a complete loss of megasporogenesis (Elliott et al, 1996), (3) In the *determinant infertile 1* (*dif1*) mutant ovule, it lacks a female gametophyte due to meiotic chromosome missegregation (Bhatt et al, 1999) even though the sporophytic tissues are present; (4) In the *inner no outer* (*ino*) mutant ovule, it lacks sporophytic outer integuments of the ovule (Villanueva et al, 1999). In each of these pistils, the number of emerged pollen tubes and that of ovules showing funicular guidance and polytubey were examined (Fig. 5). Pollen tubes elongating along the female tissues including septum surface and funiculus were defined as adhesions (Lord, 2000). Ovule showing multiple pollen tubes on the funiculus was also defined as polytubey (Fig. 5). (1) In the WT, pollen tubes emerging from the

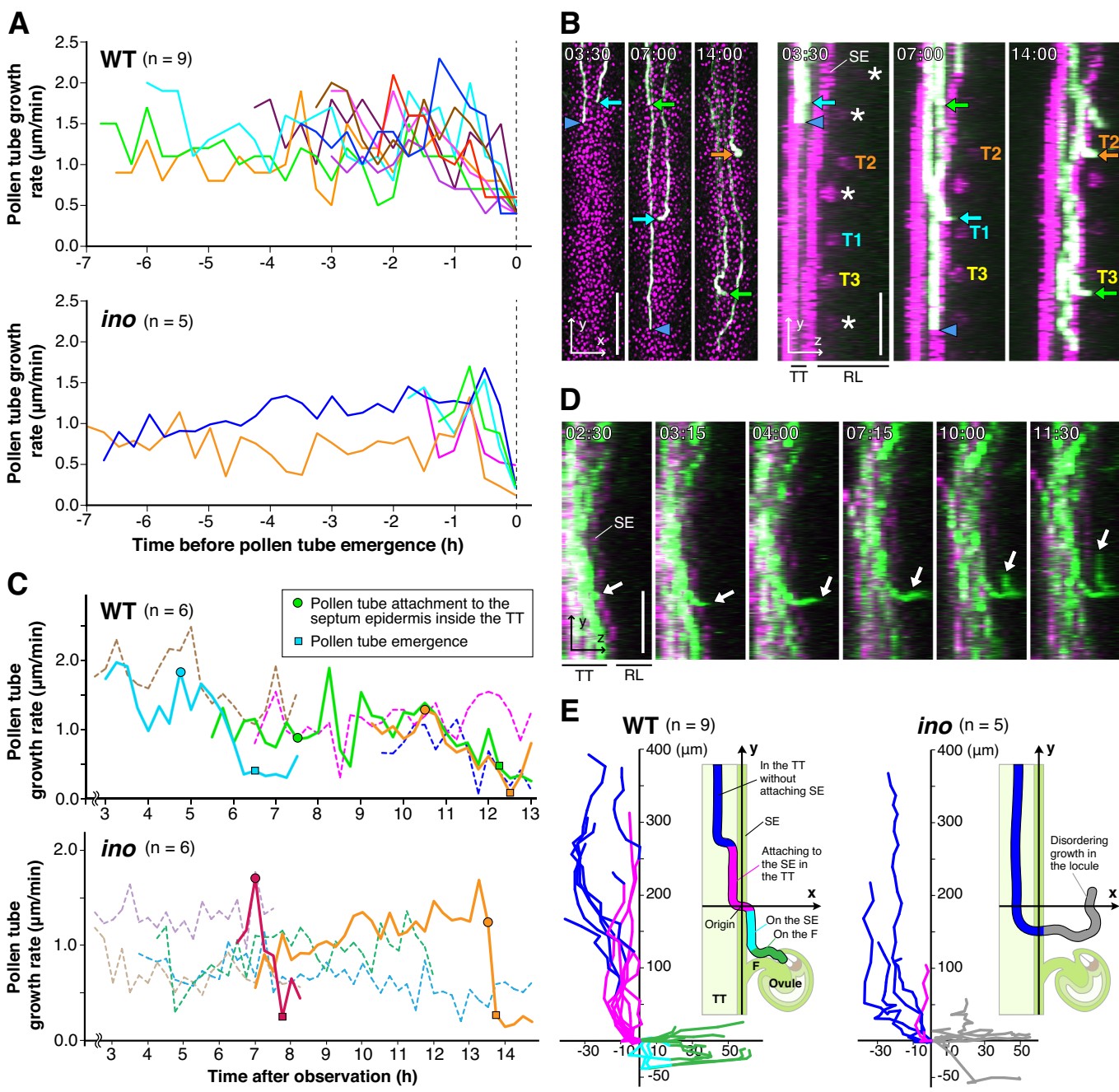

**Figure 4. Features of emerged pollen tube from the transmitting tract in the WT and *ino* mutant ovaries.**

(A) Pollen tube growth rate in the transmitting tract (TT) of 8 and 5 pollen tubes that emerged from the TT in the WT and *ino* pistils, respectively. The time of pollen tube emergence was defined as the origin. A value of *x* smaller than zero indicates a pollen tube within the TT. (B) Emergence of the three pollen tubes from the TT in the WT ovary. Asterisks show ovule autofluorescence. Arrows and arrowheads indicate emerged and non-emerged pollen tubes, respectively. Each targeted ovule is shown as T1–T3. See also Movie EV2C. (C) The pollen tube growth rate of emerged and non-emerged pollen tubes in a single pistil is shown as solid and dashed lines, respectively. (D) Pollen tube emerging from the TT of the *ino* pistil. White arrows indicate the tip of the emerged pollen tube. (E) Tracking of the tip of 9 and 5 emerged pollen tubes in the WT and *ino* pistils from the *yz*-projection images. Time-lapse imaging was performed at 15-min intervals. The emerged pollen tubes are colored by their *yz*-positions as follows: growing in the TT (blue), attaching to the septum epidermis in the TT (magenta), growing on the septum epidermis after the emergence (cyan), growing on the funiculus (green), and disordering growth in the *ino* locule without attaching SE and funiculus (gray). Data Information: (B, D) Time stamps are the same as in Fig. 1. SE septum epidermis, F funiculus, TT transmitting tract, RL remaining locule. Scale bars, 100 μm. Source data are available online for this figure.

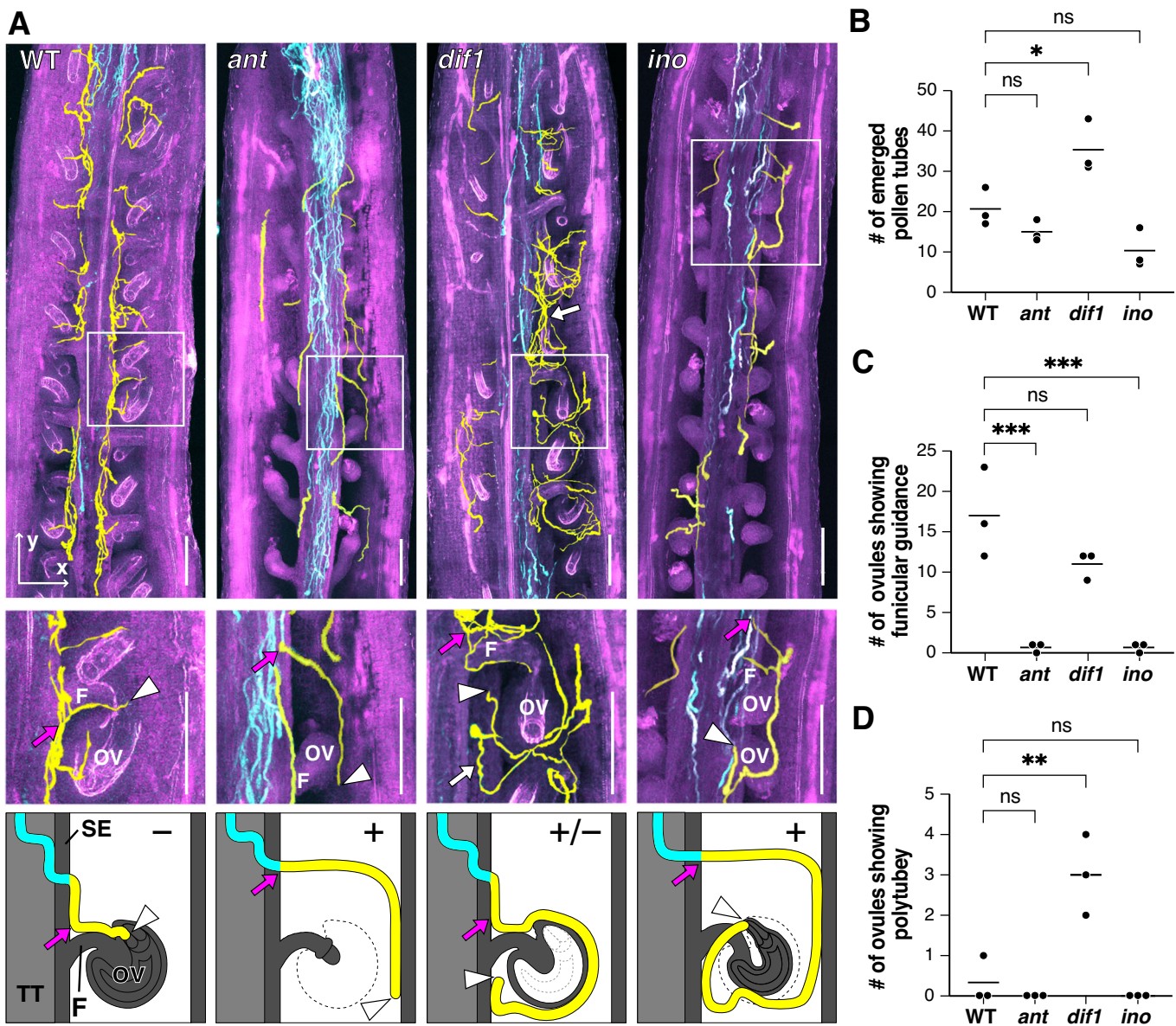

**Figure 5. Pollen tube emergence and funicular guidance in the four mutant pistils showing impaired ovular tissues.**

(A) Observation of pollen tube behavior in the ovary of ClearSee-treated maximum pollinated pistils. The pistils pollinated with pollen labeled by mTFP1 were collected 12–18 h after pollination. The xy-projection images by 8-μm steps are shown. Pollen tubes expressing mTFP1 and pistil autofluorescence are shown in cyan and magenta. The pollen tubes that emerged from the transmitting tract into the locule are colored yellow and overlaid. The magnification of the emerged pollen tube is highlighted by a white box, and its schematic representation is also shown at the bottom. Arrows and arrowheads indicate the point of the pollen tube exiting from the septum and the tip of the pollen tube, respectively. Plus and minus in the schematic representation indicate the presence or absence of pollen tube adhesion on the female tissue. Plus/minus shows less attachments to the female tissue surfaces (white arrow). F funiculus, OV ovule, SE septum epidermis, TT transmitting tract. Scale bars, 100 μm. (B–D) Features of emerged pollen tubes in the WT and three mutant ovaries. Pistils pollinated by pollen labeled with mTFP1 and Venus were collected at 6 h after pollination and cleared by ClearSee. Three pistils per mutant were averaged (n = 3 pistils). (B) Number of emerged pollen tubes. (C) The number of ovules having pollen tubes on the funiculus, which was defined as showing funicular guidance. (D) The number of ovules having multiple pollen tubes on the funiculus is defined as polytubey. Data information: In (B–D), The vertical bars show mean values. Statistical significance was determined using a one-way analysis of variance (ANOVA) followed by Dunnett's multiple comparisons test. *p < 0.05; **p < 0.01; ***p < 0.001; ns not significant. Source data are available online for this figure.

TT adhered to the septum surface, and then elongated along the funiculus toward the ovule (Fig. 5A). In the pistil at 6 HAP, 20.7 ± 4.7 pollen tubes emerged from the TT, and 17.0 ± 5.6 ovules showed funicular guidance, of which 0.3 ± 0.6 ovules showed polytubey as multiple pollen tubes on the funiculus (mean ± s.d.; n = 9 pollen tubes;

Fig. 5B–D). To confirm the requirement of sporophytic/gametophytic tissues in the ovules for pollen tube behaviors, three mutants was similarly examined. (2) In *ant* mutant ovaries, there was a similar number of emerged pollen tubes (15.0 ± 2.6 ovules, mean ± s.d., ns; Fig. 5B) but a significantly reduced number of ovules showing

funicular guidance (0.7 ± 0.6 ovules, mean ± s.d., ***p < 0.001; Fig. 5C) compared to the WT. (3) In a female gametophytic mutant *dif1*, although micropylar guidance by the ovular gametophytic cells was completely impaired, significant increase in the number of pollen tubes which emerged from the TT was observed compared to those from WT (35.3 ± 6.7 pollen tubes, mean ± s.d., *p < 0.05; Fig. 5B), whereafter many of them lost their way to the micropyle and continued their disordered growth (Fig. 5A). As a results, polytubey was significantly increased in a *dif1* ovary (3.0 ± 1.0 ovules, mean ± s.d., **p < 0.01; Fig. 5A,D). These observations suggest that ovular sporophytic cells positively regulate pollen tube emergence, whereas ovular gametophytic cells negatively regulate pollen tube emergence. Thus, we also investigated the ovular sporophytic mutant, the *ino* mutant. (4) In *ino* mutant ovaries, there was a similar number of emerged pollen tubes (10.3 ± 4.9 pollen tubes, mean ± s.d., ns; Fig. 5B), whereas a significantly reduced number of ovules showing funicular guidance (0.7 ± 0.6 ovules, mean ± s.d., ***p < 0.001; Fig. 5A,C) and that of pollen tubes attached to the septum and funiculus (Fig. 5A), suggesting that the sporophytic ovular outer integument enhances pollen tube adhesion to the maternal tissue surface. Consistently, disordered pollen tube adhesion to the septum and funicular surfaces was observed in the (2) *ant* mutant that also lacks outer integuments (Fig. 5A). Thus, we concluded that sporophytic ovular outer integument enhances pollen tube adhesion to the maternal tissue surface including septum and funiculus. The results of pollen tube emergence in the (3) *dif1* and (4) *ino* also suggest that the female gametophytic cell-derived factors negatively regulate pollen tube emergence and positively regulate polytubey block on the funiculus, independent of fertilization.

To investigate the regulation by the outer integument, pollen tube emergence in the *ino* ovary was assessed through two-photon live imaging. In the WT, pollen tubes gradually changed their growth direction within the TT, then elongated along the SE for ~50–300 µm before emergence (Fig. 4E). In contrast, pollen tubes in the *ino* TT did not grow along the SE (Fig. 4E) and emerged rapidly without decreasing their growth rate (Fig. 4A,C). After emergence from the TT, it was observed that the pollen tube in the locule elongated upward with disordered growth without growing toward the ovule (Fig. 4D). The *Arabidopsis* TT comprises longitudinally elongated cells and a large number of extracellular matrix components (Crawford et al, 2007); however, there were no obvious morphological differences between the TTs of WT and *ino* (Fig. EV4B). These results demonstrate that an unknown long-distance guidance signal(s) derived from the ovular sporophytic outer integument affects the pollen tube within the TT, which facilitates pollen tube emergence and attachment to the female tissue surfaces.

## Dissection of FER- and LRE-dependent polytubey blocks in pollen tube emergence and funiculus entry

We investigated when and where the female sporophytic or gametophytic cells regulate pollen tube emergence from the transmitting tract and polytubey block on the funiculus. As mentioned above, FER is widely expressed in female tissues, plays important roles, and is involved in polytubey block at both the septum and funiculus (Duan et al, 2020; Zhong et al, 2022). The number of ovules showing single pollen tube or multiple pollen tubes (polytubey) on the funiculus at 4, 6, and 18 HAP was analyzed by NaOH-cleared WT and *fer* pistils stained with aniline

blue solution. We also analyzed the ovules accepting homozygous *generative cell-specific 1* (*gcs1*) mutant pollen tubes as controls because polytubey in *gcs1/hapless 2* (*hap2*) pollen-pollinated WT pistil occurs by a different mechanism, known as fertilization recovery (Beale et al, 2012; Kasahara et al, 2012; Nagahara et al, 2015; Duan et al, 2020). In the *fer* mutants, one-to-one pollen tube guidance was impaired, resulting in multiple pollen tubes on the funiculus of each ovule (polytubey, Fig. 6A). It should be noted that most pollen tubes on the *fer* funiculus did not overlap as reported in the *magatama* mutant (Shimizu and Okada, 2000), but most *gcs1* pollen tubes were attracted to the same route as the first pollen tube on the funiculus, as previously reported (Kasahara et al, 2012), consistent with different mechanisms underlying the polytubey of these mutants (Fig. 6A).

We first hypothesized that FER-dependent blocking (Zhong et al, 2022) mainly regulates one-to-one pollen tube guidance at the septum. To investigate blocks during pollen tube emergence, WT and *fer* mutant pistils at 4 HAP were investigated when fertilization was incomplete in most ovules. The number of emerged pollen tubes was significantly increased in the *fer* pistil, even at 4 HAP, but not in the WT pistil (Fig. 6B). However, the position of pollen tube emergence on the septum surface (apical preference) did not differ significantly (Fig. 6C). This finding supports the notion that FER-dependent sporophytic signaling functions upon emergence onto the septum as a block by restricting the number of emerged pollen tubes.

We further investigated polytubey blocks at funicular guidance after pollen tube emergence. The distribution of ovules with multiple pollen tubes on the funiculus, defined as polytubey, was investigated in NaOH-cleared pistils. Polytubey was scarce among WT ovules but prevalent in *fer*, even at 4 HAP (Fig. 6D). We also analyzed the mutant of *LORELEI* (*LRE*) that interacts with FER in synergid cells but is not expressed in sporophytic tissues (Liu et al, 2016). In the *lre* mutant, both the number and the position of emerged pollen tubes were not significantly different from those in the WT (Fig. 6B,C), whereas polytubey was observed at high frequency at 4 HAP as well as in the *fer* mutant (Fig. 6A,D). The *gcs1* mutant and WT showed few polytubey events at 4 and 6 HAP (Fig. 6A,D). This result suggests that FER mainly controls the number of emerged pollen tubes at the septum, whereas synergid cell-dependent gametophytic signal(s) regulate the number of pollen tubes to enter into the funiculus. These multistep controls achieve one-to-one pollen tube guidance as polytubey blocks.

## Strict, multiple blocking signals are established on the funiculus 45 min after the first pollen tube enters the funiculus

A mutation in synergid cell-specific *LRE* caused polytubey at the step of funiculus entry step (Fig. 6A) but not caused increasing emerged pollen tubes (Fig. 6B). We hypothesized that the gametophytic FER- and LRE-dependent signaling regulates polytubey block on the funiculus, even before the arrival of the first pollen tube at the synergid cell that exerts the blocking signal (Duan et al, 2020). The spatiotemporal behavior of pollen tubes during polytubey was observed under the single-locule method. Live imaging results show that across 110 WT pistils from 1 to 18 HAP, 448 ovules displayed pollen tube attraction of which 18 possessed multiple pollen tubes on the funiculus, which was defined

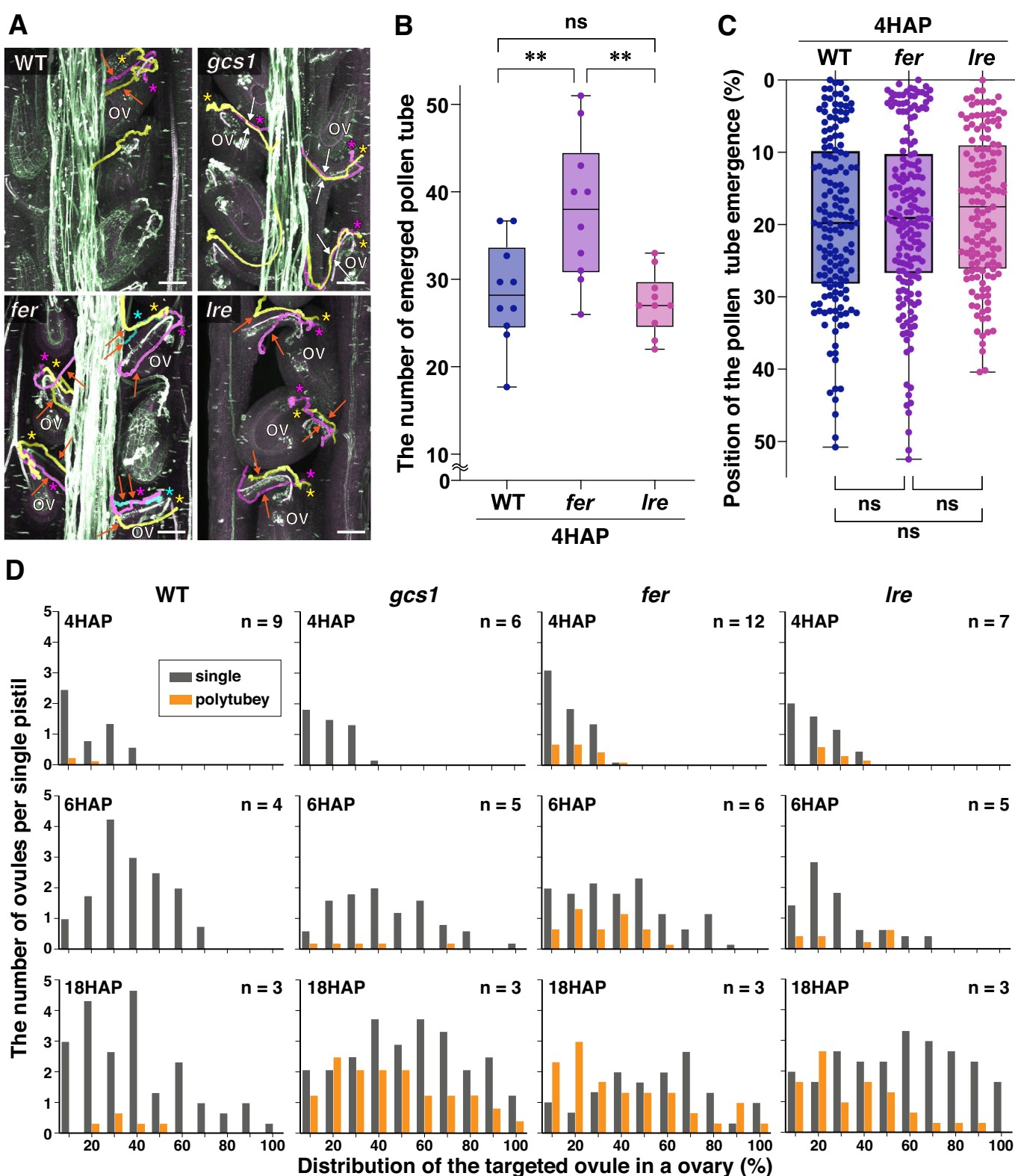

as polytubey, representing a rate of 4.0% (Fig. 7A; Movie EV3A). This ratio is comparable to that reported for polytubey in NaOH-cleared pistil (around 10% in Zhong et al, 2022; 1.6% in Capron et al, 2008). Our live imaging showed more polytubey in the *fer* (27 pistils: 111 ovules exhibiting attraction, 37 with polytubey; a rate of

33%) than in the WT (Fig. 7B,D), similar to the previous report in the NaOH-cleared pistil (over 40%, Zhong et al, 2022), and more than that in the heterozygous *fer* mutant (*fer*/+) NaOH-cleared pistil (about 10%, Huck et al, 2003). Polytubey was also frequently observed in *lre* under the single-locule method (19 pistils: 71 ovules

**Figure 6. Spatiotemporal analysis of pollen tube emergence and funicular guidance in the cross showing polytubey by using *fer*, *lre*, and *gcs1* mutants.**

(A) Funicular guidance in the ovaries. Pollen tubes in the pistil were stained with aniline blue in the NaOH-clearing solution. Multiple pollen tubes adhering on the same funiculus are colored overlay with yellow, magenta, and cyan with same-colored asterisks. Multiple pollen tubes that overlapped at the same route on the funiculus are marked with white arrows, whereas those that did not overlap on the funiculus are marked with orange arrows. OV, ovule having multiple pollen tubes on the funiculus. (B) Number of emerged pollen tubes in the pistils of WT, *fer*, and *lre* at 4 h after pollination (HAP), ten independent experiments (n = 10 pistils). (C) Position of the pollen tube emergence on the septum epidermis at 4 HAP in the pistils of WT (n = 153 pollen tubes), *fer* (n = 176 pollen tubes), and *lre* (n = 135 pollen tubes); five pistils per genotype. (D) Average number of ovules showing single pollen tube attraction and polytubey per ovary at 4, 6, and 18 HAP. The ovules showing multiple pollen tubes on the funiculus were defined as polytubey. Bar charts represent the number of ovules present in each ten percentile of ovary length from the most apical (0–10%) to the most basal (90–100%) ovules of each pistil. Data information: In (B, C), box plots represent the median with 25th and 75th percentiles with minimum and maximum whiskers. Data were presented as mean ± s.d. Statistical significance was determined using one-way analysis of variance (ANOVA) followed by Tukey's multiple comparisons tests. **p < 0.01; ns not significant. (A) Scale bars, 50 μm. Source data are available online for this figure.

displaying attraction, 35 with polytubey; a rate of 49%, Fig. 7C,D), which was more than the previous report in the NaOH-cleared pistils (17.9%, Tsukamoto et al, 2010).

Next, the timing of funicular guidance initiation between the first and second pollen tubes was also analyzed. All ten polytubey events observed in the WT (in which the second pollen tube was attracted without repulsion) were observed within 45 min (Fig. 7D). Similarly, 31 of 37 (84%) and 29 of 35 (83%) polytubey events were observed within 45 min for *lre* and *fer*, respectively. These results suggest that the polytubey block is not strictly prohibitive within 45 min after entry of the first pollen tube onto the funiculus, even in the WT. The frequent polytubey within 45 min in both *fer* and *lre* suggests that the distantly locating synergid cell is involved in rapid but non-strict blockage of entry onto the funiculus. To determine whether such blockage depends on the sporophyte or gametophyte, the number of polytubey was analyzed in the NaOH-cleared heterozygous *fer* and *lre* mutant pistils. Heterozygous *fer* and *lre* mutants showed frequent and comparable polytubey on the funiculus at 6 HAP, suggesting that it is not the sporophytic but rather the gametophytic signal that is required for blocking second pollen tube entry onto the funiculus (Fig. 7E). Notably, we observed that several pollen tubes returned from the funiculus to the septum (i.e., repulsion) in both *fer* and *lre* mutants even they were the first pollen tubes, but not in the WT (arrows in Fig. 7B,C; Movies EV3B,C). Therefore, this new repulsive signal may be inhibiting the funicular guidance of excess pollen tubes, which is still active in both the unfertilized *fer* and *lre* ovaries. There are at least two stages of polytubey blockage after the start of funicular guidance: The first stage is incomplete, and begins from the time when the first pollen tube begins funicular guidance, but stronger blockage works after 45 min as the second stage. Under our observation, the first block was defective in both the *fer* and *lre* mutants, but the second blocking mechanism was evident in both mutants, albeit incompletely. Our study reveals a repulsion mechanism at the pollen tube entry step on the funiculus, suggesting an important new aspect in the multiple polytubey blocks.

## Discussion

One-to-one pollen tube guidance is critical for effective reproduction. In this study, our live imaging suggested that ovules provide multiple signals to achieve one-to-one guidance. The first step in this process was selection within the TT, which depended on pollen tube distribution, suggesting that selection is stochastic rather than dependent on pollen tube capacitation (Sankaranarayanan and Higashiyama, 2018) and that the emergence signal(s) is not uniform throughout the TT but is more effective in the vicinity of SE (Fig. EV5).

Moreover, pollen tube emergence significantly increased in *dif1* but was normal in *ant* and *ino* mutants. Ovular outer integument-dependent signals may enhance pollen tube emergence and regulate adherence to the maternal tissue surfaces, such as the funicular and ovular surfaces and the septum epidermis, causing a decrease in growth rate and pollen tube emergence into a locule. These signals are likely to be localized to the surfaces, but they may also extend into the TT (Fig. 7F); the more mature the ovule, the deeper the signal(s) for emergence might reach into the TT. Both gametophytic signals (Wang et al, 2023) and regulatory sporophytic molecules have been associated with pollen tube guidance (Lausser et al, 2009). These signals and molecules include exogenous gamma-aminobutyric acid (Palanivelu et al, 2003; Yu et al, 2014), arabinogalactan proteins (Lora et al, 2019), ovular methyl-glucuronosyl arabinogalactan (Mizukami et al, 2016), the ethylene precursor 1-aminocyclopropane-1-carboxylic acid (Mou et al, 2020), and ovary-expressed bHLH transcription factor (Cheng et al, 2023). However, the nature of directional cues from distant ovular sporophytic tissue remains unclear. Molecules directly involved in the adhesion of pollen tubes to female tissues are undescribed, although stylar cysteine-rich adhesin (SCA) is known to enhance the attraction activity of chemocyanin and adhesion to the lily pistil (Kim et al, 2003; Park et al, 2000). Our findings may accelerate the identification of ovular outer integument-derived signaling molecules involved in pollen tube adhesion and attraction.

Polytubey block is achieved in multiple steps, and mutants lacking these steps have been studied, including *gcs1*, *fer*, *lre*, *maa1*, *maa3*, and *myb98* (Capron et al, 2008; Escobar-Restrepo et al, 2007; Kasahara et al, 2012; Kasahara et al, 2005; Shimizu and Okada, 2000; Tsukamoto et al, 2010). FER, ANJ, and HERK1 receptors on the septum interact with pollen tube-produced RALF peptide ligands, which are involved in polytubey block (Zhong et al, 2022). We revealed a blocking defect in the *fer* ovary, which exhibited a higher number of pollen tubes, although the location of their emergence remained unchanged. The polytubey block is achieved by preventing pollen tube entrance into the micropyle through multiple steps, including dispersion, modification, and degradation of pollen tube targeting chemoattractants (Duan et al, 2020; Maruyama et al, 2015; Völz et al, 2013; Yu et al, 2021). Through our novel imaging approaches, we discovered that regulation also occurs earlier, during entry on the funiculus (Fig. 7F). A block was rapidly established following the entry of the first pollen tube, dependent on gametophytic FER and LRE. In the first 45 min, this block was less restrictive, even in the WT. This result is consistent with genetic experiments showing that polyspermy in *Arabidopsis* can be explained by fertilization via more than one pollen tube (Beale et al, 2012; Grossniklaus, 2017; Mao et al, 2020; Nakel et al, 2017; Scott et al,

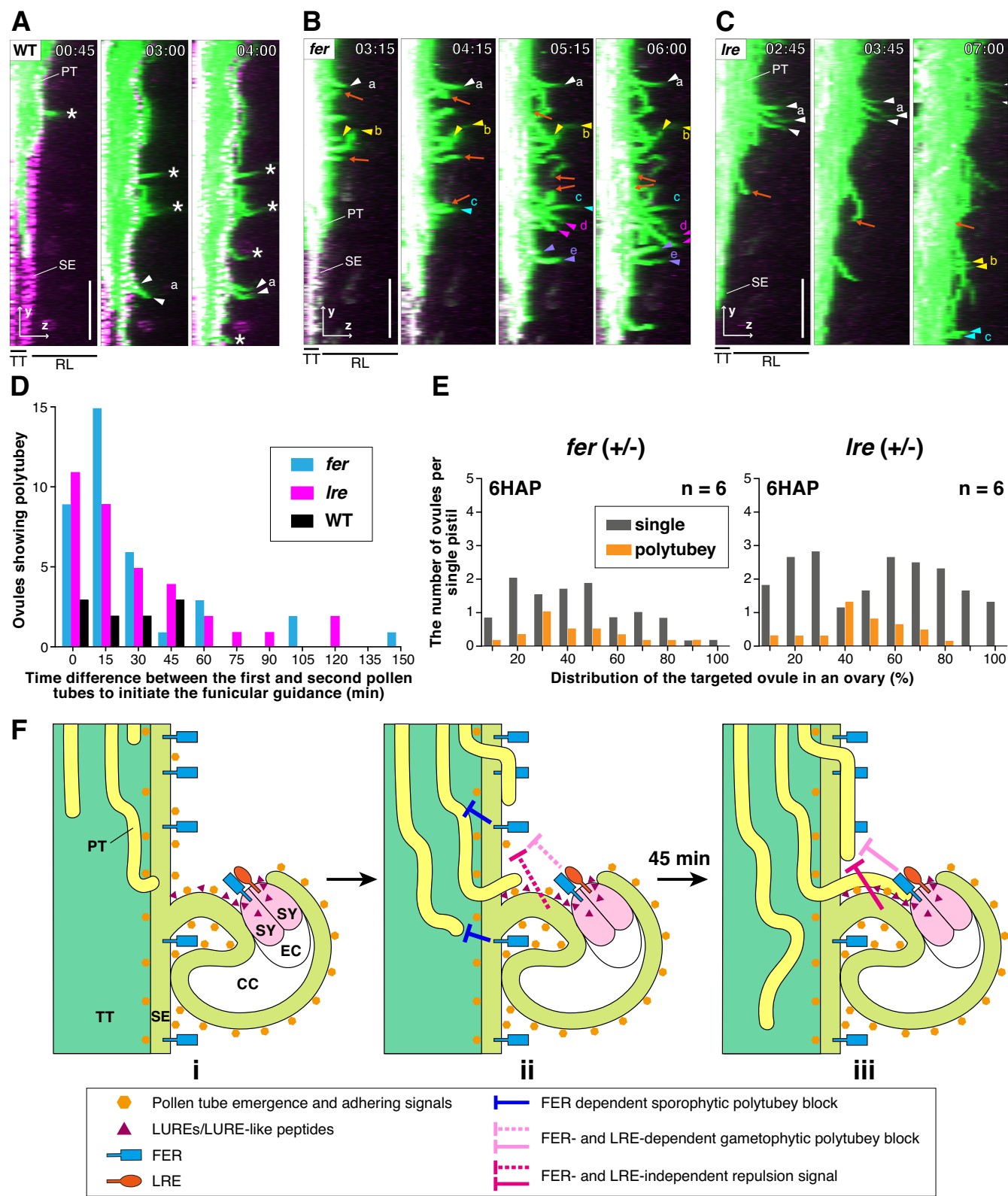

2008). Polyspermy block exists in plant gametes, although the block is more strict and rapid in the egg than in the central cell (Faure et al, 1994; Nagahara et al, 2021). On the egg cell, an almost simultaneous pollen tube discharge is required for polyspermy (Grossniklaus, 2017), but its mechanistic basis is unclear under one-to-one pollen tube guidance (Sugi and Maruyama, 2023). Our live imaging suggests that two pollen tubes simultaneously approaching the same ovule within 45 min may induce polyspermy in *Arabidopsis*.

Figure 7. Polytubey in the WT and mutant ovaries and model of one-to-one pollen tube guidance.

(A–C) Polytubey in wild-type (WT), *fer*, and *lre* pistils pollinated with mApple-expressing pollen. Polytubey was defined as multiple pollen tubes attracted to the same ovule. The *yz*-maximum projection images in 7- or 8-μm steps, including the transmitting tract, are shown. The pollen tube and pistil autofluorescence are shown in green and magenta, respectively. Asterisks indicate single pollen tube guidance. Pollen tubes on the same funiculus are shown as same-colored arrowheads. Small letters indicate ovules with multiple pollen tubes on the funiculus. Arrows indicate pollen tubes that exit the septum but return to the septum surface. Time stamps are the same as in Fig. 1 See also Movie EV3. (A) WT pistil with nuclei of the septum epidermis labeled by NLS-YFP is shown in magenta. (B) Pistil of *fer* mutant. (C) Pistils of *lre* mutants. (D) Time difference between the commencement of ascent of the first and second pollen tubes on the funiculus in the case of polytubey that was defined as two pollen tubes attracted to the same ovule. The cases of polytubey in 10 in WT, 37 in *fer* and 35 in *lre* are shown. The funicular guidance time for each first pollen tube was set at 0 min. (E) Average number of ovules showing single pollen tube attraction and polytubey per ovaries of heterozygous *fer* (+/−) and *lre* (+/−) at 6 h after pollination (HAP). The ovules showing multiple pollen tubes on the funiculus were defined as polytubey. Bar charts represent the number of ovules present in each ten percentile of ovary length from the most apical (0–10%) to the most basal (90–100%) of each pistil. (F) Model of multiple polytubey blocks spatiotemporally regulated by female sporophytic and gametophytic cells in the WT *Arabidopsis* pistil. (i) A long-distance guidance signal dependent on the ovular sporophytic outer integument affects pollen tube attachment to maternal surfaces, resulting in pollen tube emergence. (ii) FER prevents the emergence of excess pollen tubes on the septum. The emerged pollen tube elongates with attachment to the septum surface and receives funicular guidance signals, such as LUREs and LURE-like peptides, in a spatially restricted region of the funiculus. The FER- and LRE-dependent signal(s) derived from the synergid cells and the FER- and LRE-independent signal(s) regulate the polytubey block at the entrance of the funiculus, which acts as a repulsion signal for the second pollen tube. (iii) More stringent blocking signals are established at the funiculus 45 min after the entry of the first pollen tube. Data Information: (A) scale bar = 100 μm. (A–C, F) PT pollen tube, TT transmitting tract, SE septum epidermis, SY synergid cell, EC egg cell, CC central cell, RL remaining locule .Source data are available online for this figure.

The molecular properties of the polytubey block upon pollen tube entry onto the funiculus remain elusive. Multiple pollen tubes are attracted along different paths on the funiculus in the *fer* and *lre* ovaries (Fig. 6A) because the gametophytic-dependent attractive funicular guidance signal is increased and/or expanded, or the hypothetical repulsive signal decreases and/or narrows. In the synergid cell, FER-dependent signaling mediated by de-esterified pectin induces nitric oxide production, which inactivates LUREs (Duan et al, 2020). Our results suggest long-distance communication between the first pollen tube and the synergid cell because the block commenced before the first tube arrived at the micropyle (Fig. 7D,F). Molecular diffusion before and after 45 min requires further research. Our live imaging also revealed the novel repulsion process of excess pollen tubes on the funiculus (Fig. 7B,C; Movie EV3B−D); pollen tubes that failed to enter the micropyle returned to the septum and then attempted to head for another ovule in the *dif1*, *fer*, and *lre* ovaries (Fig. EV5). However, the mechanism of such pollen tube repulsion remains largely unknown (Higashiyama and Takeuchi, 2015). This repulsion mechanism is likely to be FER- and LRE-independent because it occurred in *fer* and *lre* more frequently.

In conclusion, our two-photon imaging revealed novel dynamics and spatiotemporal signaling in one-to-one pollen tube guidance. This study may accelerate the identification of signaling molecules driving this process in angiosperms, which exhibit a unique navigation system related to sexual reproduction.

# Methods

## Plant materials and growth conditions

*Arabidopsis* ecotype Columbia (*Col-0*) was used as the WT plant. *LAT52::mTFP1*, *sGFP*, *Venus*, *TagRFP*, and *mApple* (Mizuta et al, 2015), FGR8.0 (Völz et al, 2013), *MYB98p::GFP* (Kasahara et al, 2005), and *HDG11p::NLS-YFP* (Ueda et al, 2017) have been previously described. Nuclei in the septal epidermal cells of *HDG11p::NLS-YFP* are labeled with YFP. Homozygous mutant seeds of *ant* (SALK_022770), *dif1* (*rec8-1*; SALK_091193), *fer* (*fer-4*; GK-106A06) (Haruta et al. 2014), *gcs1* (SALK_135496), *ino* (*ino-4*; N6148), and *lre* (*lre-5*; CS66102), and heterozygous seeds of *fer* (*fer-4*; GK-106A06) and *lre* (*lre-5*; CS66102) were obtained from the Arabidopsis Biological Resource Center at Ohio

State University (Columbus, OH, USA) or the GABI-KAT line (Rosso et al, 2003). Seeds were germinated on agarose plates at 22 °C under 24 h light. The primers used for genotyping each mutant are listed in Table EV1. Fourteen-day-old seedlings were transferred into a mixture of vermiculite and potting compost. Seedlings were grown at 21–24 °C under long-day conditions (16 h light/8 h dark).

## Single-locule method

The single-locule method was developed for live imaging of pollen tube guidance within a living pistil (Fig. EV2). This method normalizes the optics for *Arabidopsis* two-photon live imaging (Mizuta et al, 2015) based on a previous method (Rotman et al, 2003). Fully mature stage 14 flowers were used to avoid potential pistils with immature ovules. The flowers were emasculated 18–24 h before being hand-pollinated. A frame made of a 0.2 mm thick silicone film (AsOne, Osaka, Japan) was placed on the glass bottom dish as a mold (Fig. EV2A, Setting), as previously described (Mizuta and Higashiyama, 2014). Pollen germination medium (PGM) (Boavida and McCormick, 2007) and low-melting-point agarose (NuSieve GTG agarose; Lonza Group Ltd., Basel, Switzerland) were poured into the mold. Immediately after hand pollination, a pistil was collected on a glass slide with double-sided tape. Pistils were cut open using an ophthalmic knife (MANI, Inc., Tochigi, Japan) at the valve margins (Fig. EV2A, Dissection). The remaining locule was immediately placed horizontally on the cut side (Fig. EV2A, Setting), and the flower stalk was embedded in the solid PGM (Fig. EV2A, Imaging). Liquid exuding from the solid PGM filled the bottom of the locule by capillary action. A piece of 0.2 mm thick silicone film was then placed on top of the remaining locule as a weight to prevent lifting. The silicone frame was covered with a 0.1 mm thick silicone film to maintain high humidity. Wet paper wipes were placed in a glass bottom dish to maintain humidity (Fig. EV2A, Setting), and the system was sealed with Parafilm (Bemis Flexible Packaging, Oshkosh, WI, USA).

## Two-photon imaging and data analysis

The imaging system used was based on a previous study (Mizuta et al, 2015). Images were acquired using a laser-scanning inverted

microscope (A1R MP; Nikon, Tokyo, Japan) equipped with a 25× water immersion objective lens (CFI 75 Apo 25xW MP, WD = 2 mm; NA = 1.10). A handmade water supply system was equipped with an objective lens, which enabled long-term time-lapse imaging. Emitted fluorescence signals were detected using non-descanned GaAsP PMT detectors (Nikon). To reduce plant autofluorescence and photo-damage, excitation wavelengths of 980–1000 nm were used for live imaging and 850 or 860 nm for imaging of fixed samples (Mizuta et al, 2015). Time-lapse imaging was initiated 1–2 HAP at 10- or 15-min intervals. Z-stack images were taken using multiple z-planes at 1–15-μm intervals for 3D construction, projection images, and optical sections inside the ovary. To reconstruct a large image, image stitching was automatically performed using NIS-Elements v4.10 software (Nikon). All images were analyzed with Adobe Photoshop (Adobe Systems, Inc., San Jose, CA, USA) and Fiji (Schindelin et al, 2012) software.

## Pollen tube growth rate and pollen tube distribution in the transmitting tract

Pollen tube growth rates and distribution were calculated by manually tracking the pollen tube tips on the *xy*-projected images using Fiji software. As the pistil grew under the single-locule method, an average of three random points on the SE were used to normalize the *xy*-movement associated with pistil development.

## Morphological analysis of the pollen tube and ovule inside the pistil using plant transparent regent ClearSee

Pistil clearing was performed using the ClearSee.v2 reagent as previously described (Mizuta et al, 2016; Mizuta and Tsuda, 2018). ClearSee.v2 achieves tissue transparency within the remaining spatial arrangement of organs and fluorescent signals in the *Arabidopsis* pistil (Kurihara et al, 2015). Pistils were collected and fixed with 4% paraformaldehyde (w/v) for 1 h at each time point and then cleared in a ClearSee.v2 solution for 4 weeks, according to previous reports (Mizuta and Tsuda, 2018). To assess ovule maturation, FGR8.0 immature pistils were fixed before flowering at stages 10, 11, and 12 (Smyth et al, 1990) and then cleared with ClearSee.v2. Pistils at stage 14 were emasculated at stage 12. To observe ovary tissue structure, pistils were stained with 0.1% (w/v) calcofluor staining solution after clearing with ClearSee.v2. Transparent pistils were observed using two-photon excitation microscopy (2PEM) (A1R MP, Nikon) at a suitable excitation wavelength for each fluorescent protein (Mizuta et al, 2015).

## Seed distribution under limited pollination

Stage 12 buds of WT flowers were emasculated and used for experimentation after being allowed to mature for 24 h. For hand pollination, single pollen grains were applied with tweezers to the center or side of a stigma under a stereomicroscope. After 3 d, 52 pistils with central pollination and 44 pistils with side pollination were collected. Sample fixation and clearing by ClearSee.v2 were performed in the same way as described above. After one week of clearing, transparent siliques were observed under a stereomicroscope (SZX7; Olympus, Tokyo, Japan) with an i-NTER LENS optical adapter (Microscope Network Co. Ltd.). The seed-set

distribution within the silique was analyzed using Fiji software to determine the percentile of seeds present in each of the ten percentiles of ovary length, from the most apical (1–10%) to the most basal (90–100%) aspect of each ovary.

## Targeted ovule distribution under maximum pollination

Stage 12 buds of WT flowers were emasculated and used for experimentation after being allowed to mature for 24 h. To achieve maximum pollination, stamens from the opening flower were pinched with tweezers and gently brushed across the emasculated stigma to attach pollen under a stereomicroscope. The 12 pollinated pistils were collected 6 HAP. Sample fixation and clearing by ClearSee.v2 were performed as described above. An ovule with at least one pollen tube on its funiculus was defined as the target ovule. Fiji software was used to measure the distribution of the funiculus of the targeted ovule. The percentile of seeds present in each of the ten percentiles of ovary length was determined, from the most apical (1–10%) to the most basal (90–100%) aspect of each ovary. The distribution of 205 ovules and the tips of the most growing pollen tubes in the 12 pistils were analyzed.

## Analysis of pollen tube emergence and polytubey in the pistil using transparency by sodium hydroxide with aniline blue staining

Pistils were emasculated 18–24 h prior to hand pollination. Pistils from WT, homozygous *fer* (−/−), heterozygous *fer* (+/−), homozygous *lre* (−/−), and heterozygous *lre* (+/−) plants were pollinated with WT pollen. The pollen of the homozygous *gcs1* mutant was pollinated with WT pistils. Pistils collected 4, 6, and 18 h after pollination (HAP) were fixed with a 9:1 mixture of ethanol:acetic acid and then incubated in 1 N sodium hydroxide (NaOH) for clearing (Kasahara et al, 2012). After overnight incubation, pistils were stained with 0.1% (w/v) aniline blue in a $K_3PO_4$ buffer for more than 10 min (Kobayasi and Atsuta, 2010). The stained pistils were rinsed with sterilized distilled water and then observed using two-photon excitation microscopy (2PEM) at an excitation wavelength of 850 nm (A1R MP, Nikon). The distributions of the ovules showing either single pollen tube attraction or polytubey were analyzed manually with Fiji software.

## Statistical analyses

The visualization and statistical analyses of data were performed using GraphPad Prism v9.0 (GraphPad Software, San Diego, CA, USA). Data were analyzed using one-way analysis of variance (ANOVA), followed by Dunnett's multiple comparisons tests or Tukey's multiple comparisons tests to analyze the difference in the number of emerged pollen tubes, funicular guidance, ovules with polytubey, and the location of pollen tube emergence.

## Scanning electron microscopy of the pistil

The WT pistil of *A. thaliana* was cut open using an ophthalmic knife (MANI, Inc., Tochigi, Japan) at the valve margins to expose the inner ovules. The pistil was observed with a scanning electron microscope (VHX-D500; Keyence, Osaka, Japan).

## Data availability

The microscopy images are available in the BioImage Archive database (https://www.ebi.ac.uk/bioimage-archive/) under accession number S-BIAD1104.

The source data of this paper are collected in the following database record: biostudies:S-SCDT-10_1038-S44319-024-00151-4.

## Peer review information

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

## Acknowledgements

We thank Dr. F. Berger and Dr. U. Grossniklaus for their helpful suggestions and discussion. We thank Dr. R. Groß-Hardt, Dr. R. Palanivelu, Dr. R.D. Kasahara, Dr. M. Ueda, Dr. D. Maruyama, and Dr. D. Susaki for providing plant materials and plasmids, Dr. H. Takeuchi and Dr. Y. Hamamura for providing plasmids, and S. Nasu, T. Nishii, and T. Shinagawa for assistance in preparing the materials. The seeds of *A. thaliana* mutants were obtained from the Arabidopsis Biological Resource Center at Ohio State University (Columbus, OH, USA) and the GABI-KAT line (Rosso et al, 2003). Microscopy was conducted at the WPI-ITbM of Nagoya University and supported by the Advanced Bioimaging Support through MEXT/JSPS KAKENHI (22H04926). This work was supported by grants from the Japan Science and Technology Agency (ERATO Grant no. JPMJER1004 to T.H., CREST Grant no. JPMJCR20E5 to T.H. and Y.M. and FOREST Program Grant no. JPMJFR204T to D.K.); the Japan Society for the Promotion of Science (nos. 18K14741, 20H05778 and 20H05779 to Y.M., no. 22H04668 to D.K., and nos. 16H06465 and 22H04980

to T.H.); the Program for Promoting the Enhancement of Research Universities (2022 to Y.M.); the Ohsumi Frontier Science Foundation (2023 to Y.M.); and the Sumitomo Basic Science Research Projects (2023 to Y.M.).

## Author contributions

**Yoko Mizuta**: Conceptualization; Supervision; Funding acquisition; Investigation; Writing—original draft. **Daigo Sakakibara**: Investigation; Methodology. **Shiori Nagahara**: Investigation; Writing—review and editing. **Ikuma Kaneshiro**: Investigation. **Takuya T Nagae**: Methodology; Writing—review and editing. **Daisuke Kurihara**: Funding acquisition; Writing—review and editing. **Tetsuya Higashiyama**: Conceptualization; Funding acquisition; Writing—original draft; Writing—review and editing.

Source data underlying figure panels in this paper may have individual authorship assigned. Where available, figure panel/source data authorship is listed in the following database record: biostudies:S-SCDT-10_1038-S44319-024-00151-4.

## Disclosure and competing interests statement

The authors declare no competing interests.

# Expanded View Figures

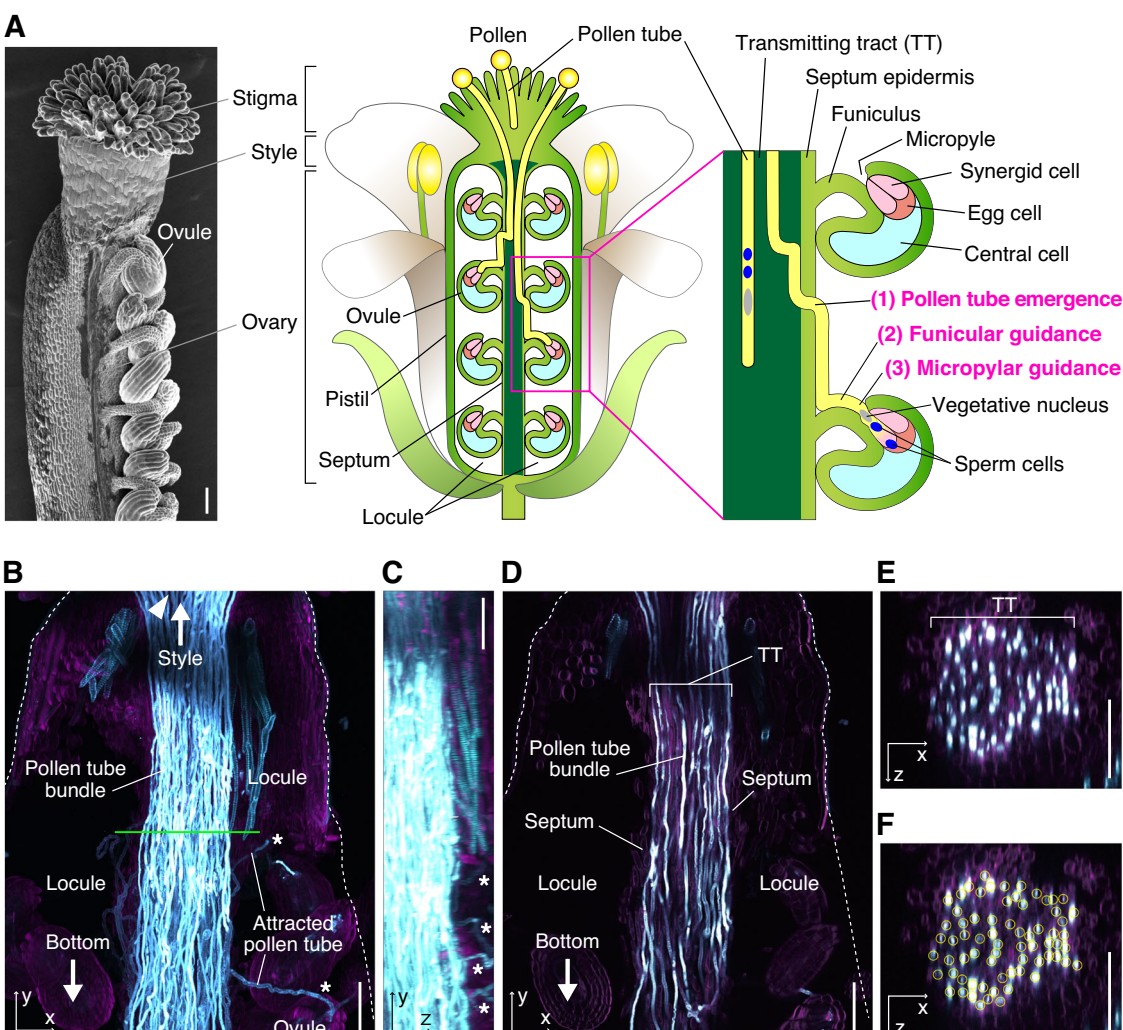

**Figure EV1. Reproductive organ structures and processes of *Arabidopsis thaliana*.**

(A) (left) Scanning electron microscope image of *A. thaliana* pistil. The right side of the ovary wall has been removed to expose the inner ovules. (right) Schematic representation of the structures of the *A. thaliana* flower and the process of pollen tube guidance inside the ovary. A flower has one pistil in the center, which is formed by fusing the carpels. The fused carpels form two locules, which harbor around 20–30 ovules each in a vertical arrangement inside the locule. Merged region forms a septum harboring the placentae, from where a stalk-like structure funiculus connects each ovule. When pollen lands on the stigma, pollen germinates in the pollen tube. Pollen tube penetrates inside the stigma and enters the style connected to the transmitting tract (TT) in the ovary. Pollen tube guidance after entering the TT was divided into three steps in this study: (1) pollen tube emergence from the TT into a locule (pollen tube emergence), (2) pollen tube guidance from the surface of the septum to the funiculus (funicular guidance), and (3) pollen tube guidance from the funiculus to the micropyle (micropylar guidance). After micropylar guidance, the pollen tube enters the micropyle of the ovule and releases sperm cells in the synergid cells, and then double fertilization with the egg cell and central cell occurs. (B–F) Pollen tubes in the TT. Wild-type pistil pollinated with wild-type pollen collected at 18 h after pollination was fixed and cleared by 1 N sodium hydroxide. Pollen tubes in the pistil were stained with aniline blue. (B) The *xy*-maximum projection image of the ovary in the maximum pollination. Fluorescent signals of pollen tubes and ovary are shown as cyan and magenta. (C) The *yz*-maximum projection image of (B). (D) Vertical optical section of (B). Pollen tubes inside the TT of the optical *xy* section are shown. (E) Optical cross-section of (B) generated by 1-μm steps with 123 planes. The location of the cross-section is indicated by a green line in (B). (F) Pollen tubes in the TT of (E) were counted by the multi-point tool in the Fiji software. The 69 pollen tubes in the TT are shown as yellow circles. Data Information: (B–D) Arrows indicate the directions of style and bottom of the pistil. Arrowhead indicates the end of the style. (B, C) Asterisks show the attracted pollen tubes to the ovule located in the back of pollen tube bundle. (B, D) White dotted lines show the ovary wall. TT, transmitting tract. (B–F) Scale bars, 50 μm.

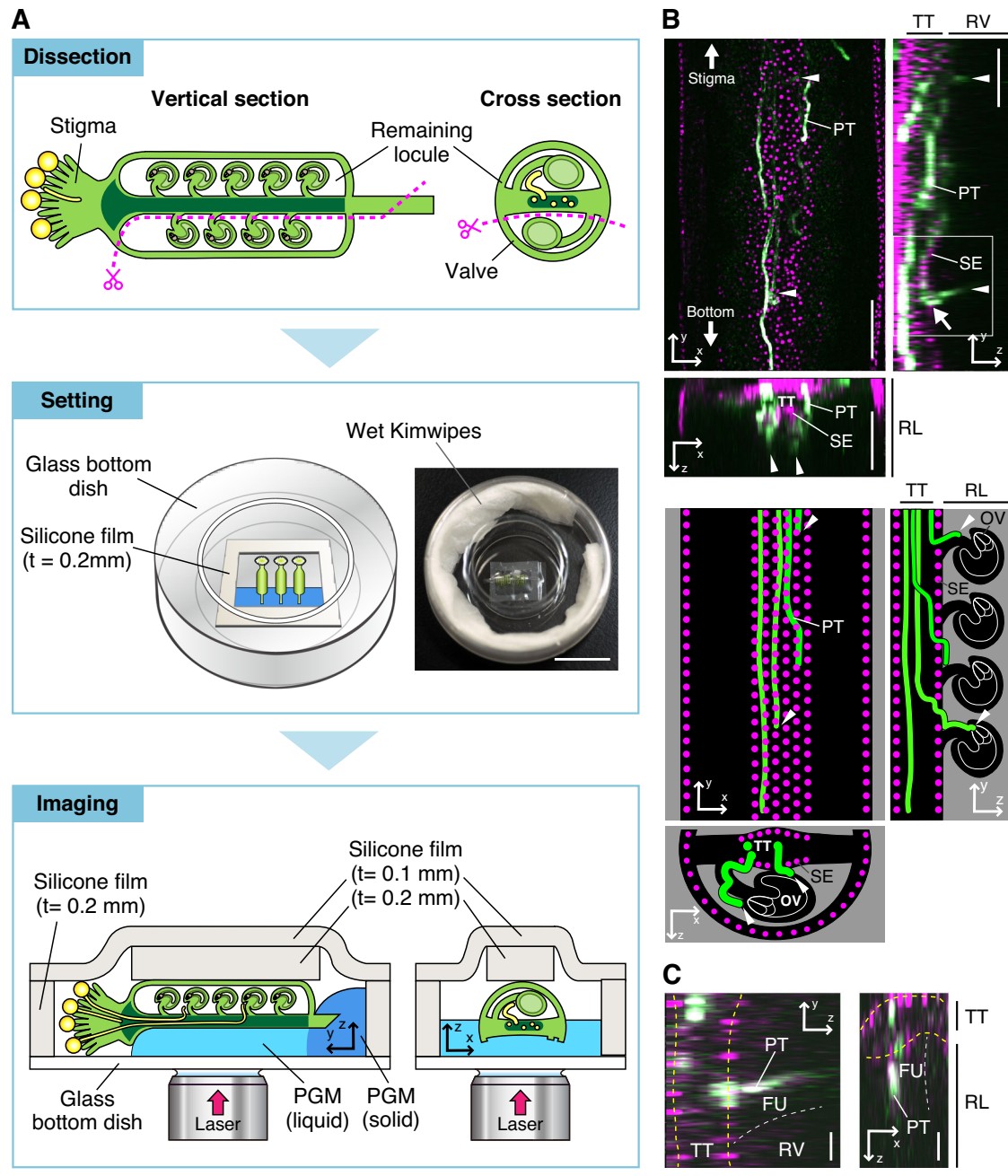

**Figure EV2. The single-locule method.**

(A) Schematic representation of sample preparation of the single-locule method. Pollinated pistils were cut and placed on the mold made with solid pollen germination medium (PGM) in the silicone film at the center of the glass bottom dish. One locule was removed without injuring the septum surface to observe pollen tube guidance inside the transmitting tract (Dissection). To maintain high humidity, wet Kimwipes were placed in the glass bottom dish, and sealed with Parafilm (Setting). These procedures must be carried out quickly and under constant temperature control to prevent tissue damage. A pistil was placed horizontally on the PGM in a silicone frame for observation by an inverted microscope (Imaging). The bottom side with the locule removed was filled with liquid PGM by capillary action. To prevent the pistils from moving and drying, silicone films were placed on both the pistil and silicone flame. Two-photon imaging was performed by the direction of the removed locule. (B) Live imaging of the pollinated pistil by two-photon microscopy under the single-locule method. The xy-, xz-, and yz-projection images are shown. Pistil from *HDG11p::NLS-YFP* was pollinated with mTFP1 expressing pollen. Epidermal nuclei and pollen tube in an ovary were labeled with YFP (magenta) and mTFP1 (green), respectively. The top is the stigma side. Schematic representation is also shown at the bottom. Arrowheads indicate attracted pollen tubes toward the ovule. The arrow indicates the point of pollen tube emergence. (C) yz- and xz-optical slice images are shown of (B) (white boxes). Fluorescent signals derived from the septum epidermis and funicular autofluorescence are shown as yellow and white dotted lines, respectively. PT pollen tube, OV ovule, SE septum epidermis, RL remaining locule, TT transmitting tract, FU funiculus. Data Information: Scale bars, 1 cm (A), 100 μm (B), and 20 μm (C).

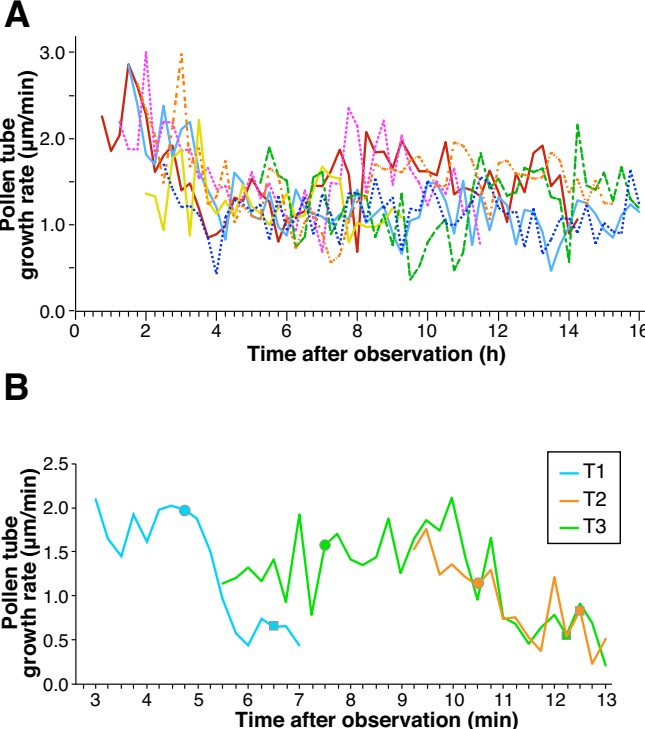

**Figure EV3. Pollen tube growth rate in the wild-type (WT) pistils under the single-locule method.**

(A) Pollen tube growth rate in the transmitting tract (TT) of the maximum pollinated WT pistil. The pollen tube growth rate of 7 non-emerged pollen tubes in a TT shown in Fig. 1B. Maximum intensity *xy*-projections with images taken at 15-min intervals were used for the analysis. See also Movie EV1A. (B) The growth rate of three emerged pollen tubes in a TT of the WT pistil with limited pollination shown in Fig. 3B. Maximum intensity *yz*-projections with images taken at 15-min intervals were used for analysis. Filled circles and squares show the time points of the pollen tube attachment to the SE and that of pollen tube emergence, respectively. See also Fig. 3C and Movie EV2C.

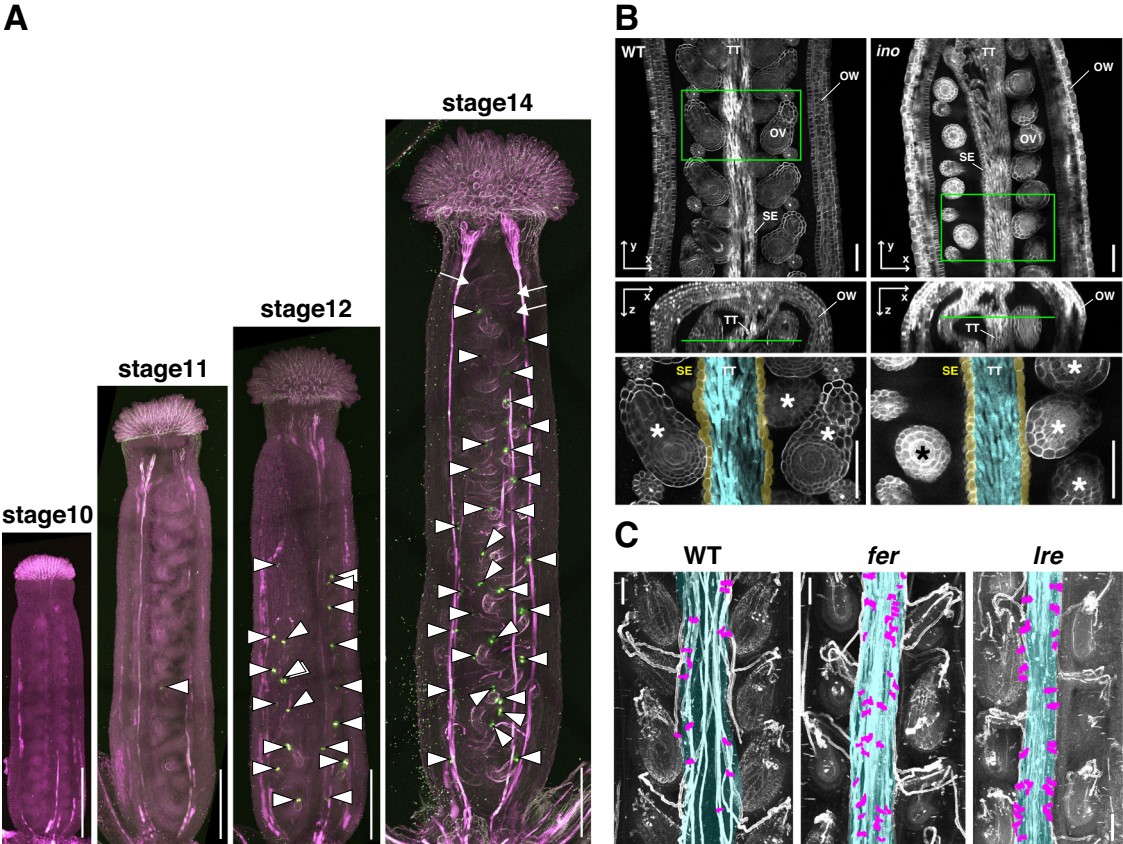

**Figure EV4. The internal structure of a transparent pistil.**

(**A**) Floral stage-dependent ovule development in the wild-type (WT) pistil. Pistils from stage 10 with petals reaching the length of the lateral stamens to stages 13–14 with opening flower (Smyth et al, 1990) were analyzed. Unpollinated pistils from *FGR8.0* were cleared by ClearSee and observed by two-photon excitation microscopy (2PEM) with 960 nm excitation. Maximum intensity projections for *xy*-projection images were generated from 25–37 z-stack images with 10-μm intervals. Autofluorescence of the pistil is shown in magenta. Arrowheads show the GFP expression in the synergid cells driven by the *MYB98* promoter of the *FGR8.0* construct. Arrows show the ovule without GFP signal in the synergid cells. (**B**) Cell wall-stained clearing WT and *ino* unpollinated pistils. Optical *xy*- and *xz*-sections were generated by 2-μm steps with 101 planes. Magnified images (green box/line) are shown at the bottom. Septum epidermis and transmitting tract (TT) are colored overlay with yellow and cyan. (**C**) Pollen tube emerging points in the WT, *feronia* (*fer*), and *lre* mutant ovaries at 24 h after pollination (HAP). Pollen tubes stained with aniline blue dye and pistil autofluorescence are shown in cyan and gray, respectively. The emerging point on the septum epidermis of each pollen tube is shown as magenta on the *xy*-projection images. SE septum epidermis, TT transmitting tract, OV ovule, OW ovary wall. Scale bars, 200 μm (**A**), and 50 μm (**B, C**).

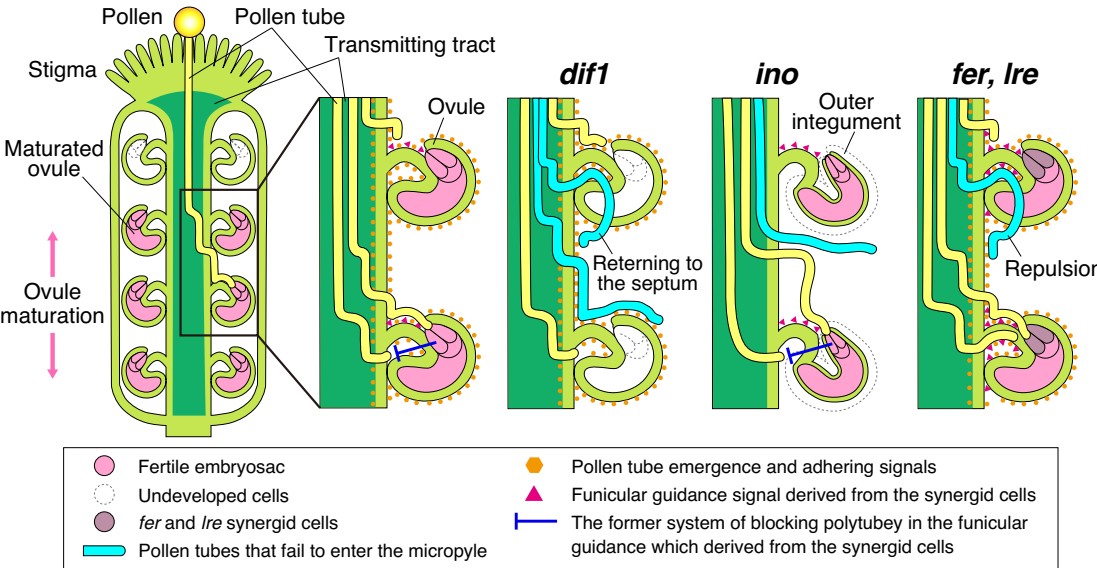

**Figure EV5. Model of one-to-one pollen tube guidance in the mutant ovaries.**

When the number of pollen tubes is limited, they are preferentially attracted to the ovule located in the lower center of the ovary, depending on the ovule maturation. The pollen tube emergence signal was derived from ovular sporophytic cells. This signal affects the attachment of pollen tubes to the maternal surfaces, which causes a decrease in growth rate and pollen tube emergence into a locule. The emerged pollen tube elongates with attachment to the septum surface and receives funicular guidance signals in a spatially restricted region of the funiculus. The signals derived from the sporophytic and gametophytic cells prevent multiple pollen tube attractions (polytubey blocks) in funicular guidance. In the *dif1* ovary, the pollen tube emergence signal from the ovular outer integument was normal, whereas the gametophytic cell-dependent polytubey block was impaired. In the *ino* ovary, which lacks pollen tube emergence and adhesion signals, a few lucky floating pollen tubes arrive at the micropyle because the signals derived from gametophytic cells function normally. Multiple pollen tubes are attracted along different paths on the funiculus in the *fer* and *lre* ovaries because the gametophytic-dependent attractive funicular guidance signal is increased and/or expanded, or the hypothetical repulsive signal decreases and/or narrows. Pollen tubes that failed to enter the micropyle returned to the septum in the *dif1*, *fer*, and *lre* ovaries. Signals derived from gametophytic cells were impaired in *dif1*, *fer*, and *lre* mutants, which causes polytubey.

