## [Peer Review File · EMBO Reports]

Deep imaging reveals dynamics and signaling in one-to-one pollen tube guidance

Yoko Mizuta, Daigo Sakakibara, Shiori Nagahara, Ikuma Kaneshiro, Takuya Nagae, Daisuke Kurihara, and Tetsuya Higashiyama

Corresponding author(s): Yoko Mizuta (mizuta.yoko.u6@f.mail.nagoya-u.ac.jp)

Review Timeline:

Submission Date:	7th Sep 23
Editorial Decision:	10th Nov 23
Revision Received:	7th Feb 24
Editorial Decision:	11th Mar 24
Revision Received:	9th Apr 24
Accepted:	18th Apr 24

Editor: *Martina Rembold*

Transaction Report:

Dear Dr. Mizuta

Thank you for the submission of your research manuscript to our journal. I apologize for the delay but we have now received the full set of referee reports that is copied below.

As you will see, all three referees consider your findings of interest to the community and overall well supported by the data, but they also raise a few concerns and have suggestions how to further improve your study that need to be addressed.

Given these constructive comments, we would like to invite you to revise your manuscript with the understanding that the referee concerns (as detailed above and in their reports) must be fully addressed and their suggestions taken on board. Please address all referee concerns in a complete point-by-point response. Acceptance of the manuscript will depend on a positive outcome of a second round of review. It is EMBO Reports policy to allow a single round of revision only and acceptance or rejection of the manuscript will therefore depend on the completeness of your responses included in the next, final version of the manuscript.

We realize that it is difficult to revise to a specific deadline. In the interest of protecting the conceptual advance provided by the work, we recommend a revision within 3 months (February 10th, 2024). Please discuss the revision progress ahead of this time with the editor if you require more time to complete the revisions.

I am also happy to discuss the revision further via e-mail or a video call, if you wish.

A few specific points:

- Table S1 should be renamed to Table EV1.
- Please provide the movie legends as simple README.txt file and then zip the legend with its movie. The zip file is uploaded.
- The nomenclature for Expanded View figures is Figure EV#. Please change the names of Figure S1-S5 accordingly.

*****IMPORTANT NOTE:

We perform an initial quality control of all revised manuscripts before re-review. Your manuscript will FAIL this control and the handling will be delayed IN CASE the following APPLIES:

- 1) A data availability section providing access to data deposited in public databases is missing. If you have not deposited any data, please add a sentence to the data availability section that explains that.
- 2) Your manuscript contains statistics and error bars based on $n=2$. Please use scatter blots in these cases. No statistics should be calculated if $n=2$.

When submitting your revised manuscript, please carefully review the instructions that follow below. Failure to include requested items will delay the evaluation of your revision. *****

2) individual production quality figure files as .eps, .tif, .jpg (one file per figure).

Please download our Figure Preparation Guidelines (figure preparation pdf) from our Author Guidelines pages <https://www.embopress.org/page/journal/14693178/authorguide> for more info on how to prepare your figures.

4) a complete author checklist, which you can download from our author guidelines

(<<https://www.embopress.org/page/journal/14693178/authorguide>>). Please insert information in the checklist that is also reflected in the manuscript. The completed author checklist will also be part of the RPF.

5) Please note that all corresponding authors are required to supply an ORCID ID for their name upon submission of a revised manuscript (<<https://orcid.org/>>). Please find instructions on how to link your ORCID ID to your account in our manuscript tracking system in our Author guidelines

(<<https://www.embopress.org/page/journal/14693178/authorguide#authorshipguidelines>>)

6) We replaced Supplementary Information with Expanded View (EV) Figures and Tables that are collapsible/expandable online. A maximum of 5 EV Figures can be typeset. EV Figures should be cited as "Figure EV1, Figure EV2" etc... in the text and their respective legends should be included in the main text after the legends of regular figures.

7) Please note that a Data Availability section at the end of Materials and Methods is now mandatory. In case you have no data that requires deposition in a public database, please state so instead of refereeing to the database. See also <<https://www.embopress.org/page/journal/14693178/authorguide#dataavailability>>. Please note that the Data Availability Section is restricted to new primary data that are part of this study.

Additional information on source data and instruction on how to label the files are available <<https://www.embopress.org/page/journal/14693178/authorguide#sourcedata>>.

10) Figure legends and data quantification:
The following points must be specified in each figure legend:

- the name of the statistical test used to generate error bars and P values,
 - the number (n) of independent experiments (please specify technical or biological replicates) underlying each data point,
 - the nature of the bars and error bars (s.d., s.e.m.)
- If the data are obtained from n {less than or equal to} 5, show the individual data points in addition to the SD or SEM.
- If the data are obtained from n {less than or equal to} 2, use scatter blots showing the individual data points.

See also the guidelines for figure legend preparation:
<https://www.embopress.org/page/journal/14693178/authorguide#figureformat>

11) Our journal encourages inclusion of *data citations in the reference list* to directly cite datasets that were re-used and obtained from public databases. Data citations in the article text are distinct from normal bibliographical citations and should directly link to the database records from which the data can be accessed. In the main text, data citations are formatted as follows: "Data ref: Smith et al, 2001" or "Data ref: NCBI Sequence Read Archive PRJNA342805, 2017". In the Reference list, data citations must be labeled with "[DATASET]". A data reference must provide the database name, accession number/identifiers and a resolvable link to the landing page from which the data can be accessed at the end of the reference. Further instructions are available at <<https://www.embopress.org/page/journal/14693178/authorguide#referencesformat>>.

12) All Materials and Methods need to be described in the main text. We would encourage you to use 'Structured Methods', our new Materials and Methods format. According to this format, the Materials and Methods section should include a Reagents and Tools Table (listing key reagents, experimental models, software and relevant equipment and including their sources and relevant identifiers) followed by a Methods and Protocols section in which we encourage the authors to describe their methods using a step-by-step protocol format with bullet points, to facilitate the adoption of the methodologies across labs.

More information on how to adhere to this format as well as downloadable templates (.doc or .xls) for the Reagents and Tools Table can be found in our author guidelines: <

<https://www.embopress.org/page/journal/14693178/authorguide#manuscriptpreparation>>.

<<https://www.embopress.org/doi/10.15252/msb.20178071>>.

13) As part of the EMBO publication's Transparent Editorial Process, EMBO Reports publishes online a Review Process File to accompany accepted manuscripts. This File will be published in conjunction with your paper and will include the referee reports, your point-by-point response and all pertinent correspondence relating to the manuscript.

Yours sincerely,

Referee #1:

In this article, the authors have used a novel 2D photon live imaging system to directly observe pollen tube behavior. Especially their growth in the transmitting tract, and pollen tube emergence into the ovary. And then how they interact with ovules. 2021). However, since we do not understand the real-time behavior of pollen tube attraction and the temporal relationships between pollen tube and various portions of the pistil, we do not understand how the pollen tubey behaviors are blocked.

Their imaging approach has revealed many new observations - seemingly small to major ones. For example, they have produced the first estimate or enumeration of the number of pollen tubes that grow in a style in vivo. This is small in the larger scheme of things, but the field has operated for a while without accurately knowing this basic information. For major findings, by using a variety of mutants in this assay, they have revealed a repulsion mechanism that is functioning specifically at the pollen tube entry step on the funiculus.

I have two broad feedbacks on this manuscript. The authors should cite major and original findings made by others in the field; I have tried to indicate where possible. Few examples:

- a. When referring to Two-Photon live imaging of pistils, the authors should mention a similar assay being done on the pistils by Rotman et al 2003
- b. Page 14 last sentence: Beale et al 2012 should be cited
- c. Line 4, Page 20: Tsukamoto et al 2010 should also be cited
- d. Line 1, Page 21, Beale et al 2012 should be cited.

Second, the writing is not great in many places and they almost provide no explanation or leave out crucial details and as a result, the complex findings and issues are not clearly written. I suggest they make a good-faith effort to elaborate on certain sections to articulate their findings even better.

Specific points (arranged by page number and not based on importance). Unfortunately, the journal-provided text file does not have line numbers so I am using manually counted line numbers within a page and the page number on the document to provide the feedback.

Line 12, Page 4: 'Arabidopsis thaliana flowers possess single ovaries; is not correct. How can it be single ovaries? Best to describe that the Arabidopsis pistil consists of two fused carpels. Each carpel represents one locule of the ovary that contains many ovules.

Line 10, Page 5: regulate is an extra word there and needs to be dropped.

Line 10, Page 5: FERONIA is misspelled with two R; one R needs to be dropped.

Lines 2 and 3, Page 7: Data and conclusions based on that data are being stated. Hence, please reference this conclusion to a figure in the manuscript.

Line 10, Page 7: 'Some pollen tube abruptly separated': How many times this was observed? Otherwise, are they concluding based on the observations of a single pollen tube?

Line 10, Page 7: Pollen tube bundle: It would enhance the understanding (especially given the experiments and results in Figure 2), if it is indicated (i) where is the pollen tube emergence site by noting the distance of emergence site from the stigma and (ii) whether this pollen tube was part of the pollen tube bundle or another tube that is behind the pollen tube bundle front that emerged when the pollen tube front has long crossed this pollen tube emergence site.

Line 16, Page 7: Should be rewritten as 'this is the first report of such a dynamic recording of the entire guidance of a single pollen tube.

Line 8, Page 9: Starting from "Pollen tube behavior and growth path". From this point onwards, it would help to have this be part of a separate paragraph. Otherwise, it is a really long paragraph with multiple results/outcomes being discussed.

Line 11, Page 9: About the pollen tube interaction with SE: How many times this was observed? Otherwise, are they concluding based on the observations of a single pollen tube?

Line 4, Page 10: Figure 4 A reference: What is the number of emerged pollen tubes from which this conclusion is based? Is it 11 tubes, as mentioned below?

Page 10: After the authors mention the results on pollen tubes that bind SE (Figure 4B-C and Figure S3B) and have not said anything, I am left wondering what happens to the growth rate of the tubes in wild-type pistils after they emerge. Why not describe it here so that there is a comprehensive and complete understanding of what happens to the tubes in the wild type? Instead of breaking this apart like what was done here, and the remainder be shown later after describing the results in Figure 5 (first few sentences at the top of page 14)?

Page 12: In this whole paragraph, for each genotype, it might be better to mention what they have and what they lack, as the goal is to discern the individual roles of sporophytic and gametophytic tissues. For example,

1. WT has both sporophytic and gametophytic tissues in an ovule.
2. In ant, both sporophyte and gametophyte are severely affected.
3. In dif1, they lack gametophyte even though the sporophytic tissues are present.
4. In the ino mutant, they lack outer integument of the ovule.

Stating this way helps when concluding that since pollen tube emergence was observed in 1 and 3, sporophytic tissues play a key role in pollen tube emergence from the TT. Without a clear description as to what each genotype lacks and contains, it is not clear why the conclusion was drawn based on the observations.

The penultimate sentence (2nd from the bottom) on Page 13: Stating that 'pollen tube emergence in the ino ovary was assessed through two-photon live imaging" confused me. Up until this point in this section, I assumed that the results shown in Figure 5 were all based on two-photon live imaging. However, the authors mention here that they used two-photon live imaging only to confirm the results in Figure 5. Upon reading the figure legend, it seems the observations were made with cleared pistils. Therefore, the authors should make a special mention of what technique was used when describing the results shown in Figures 5 and 4.

Line 3, Page 16: NaOH-cleared pistils: Make a distinction like this throughout the manuscript. In the text when describing the experiments involving each mutant, they should mention, whether the analysis was using cleared pistils or two-photon live imaging

The fourth line from the bottom of Page 16: Since FER at 4 HAP (but not LRE mutant) showed that more tubes are emerging from the TT into the ovary, how about stating that FER mainly controls the polytubey at the time of pollen tube emergence in the septum (like the way they said they are testing this hypothesis at the beginning of this paragraph)?

Page 16, section title (3 sentences from the bottom): The section title sentence is incomplete. More words following the words 'pollen tube enters' must be added:

"....the ovary or first pollen tube emergence"

Line 5, Page 17: How do these rates compare to the in vivo rates of polytubey in the first reports made on these mutants (Huck et al 2003, Tsukamoto et al 2010)? The authors should cite those numbers to both show how well their observations compare with those and also to cite those references for making these observations in vivo.

Page 18: "The extent of the polytubey block became stricter 45 min after the entry of the first pollen tube"
Could there be an alternate explanation for pollen tubey block becoming stricter? Since there are only ~70 tubes (their own data at the beginning of the manuscript and there are not enough tubes to get more than two tubes per ovule and therefore it appears as if the block is intensifying after the entry of the first pollen tube. Hence this sentence should be changed to simply reflect the observation rather than imply that the block became stricter.

Figure 4C. The word 'observation' is misspelled on the X-axis of the graph

Referee #2:

The manuscript entitled 'Deep imaging reveals dynamics and signaling in one-to-one pollen tube guidance' by Yoko Mizuta et al. reported the study on the functional mechanisms underlying the directional cues and polytubey blocks in the depths of the pistil. The authors developed a "single-locule method" using two-photon excitation microscopy to perform deep imaging of pollen tube guidance in living ovaries of *A. thaliana*. Using novel imaging methods, the authors found that the emergence of pollen tubes is stochastically determined by the distribution of pollen tubes. Mutant studies involving *dif*, *ant* and *ino* revealed that ovular outer integument-dependent signals may enhance pollen tube emergence and ovule maturation contributes to ovule targeting. Further results demonstrated that FER-dependent sporophytic signaling as a polytubey block by restricting the number of emerged pollen tubes onto the septum. Additionally, gametophytic FER- and LRE-dependent signaling regulates polytubey block on the funiculus, even before the arrival of the first pollen tube at the synergid cell that exerts the blocking signal. The author also revealed FERONIA- and LORELEI-independent repulsion signal involved in polytubey blocks on the ovular funiculus. The authors conclude that multiple signals from the ovules are involved in achieving one-to-one guidance, with polytubule blocks occurring through multiple steps. The methodology established by the authors for the investigation of in-vivo pollen tube guidance holds significant implications for advancing research in this area. Therefore, it is of general interest and worthy of publication in EMBO Reports.

Comments:

1. In Figure 4E, the emerged pollen tubes can grow on the funiculus in *ino* pistils, but there was a significantly reduced number of ovules showing funicular guidance in Figures 5A and 5B for *ino* mutant ovaries. It is necessary for the authors to discuss the possible reasons for this geographical difference.
2. In Page 12, the author mentioned that "In *ant* mutant ovaries, there was a reduced number of emerged pollen tubes and ovules showing funicular guidance" this conclusion is not well reflected in Figure 5B. I advise that the authors confirm whether the statement is correct.
3. The experimental steps of two-photon imaging, especially the post-processing of the imaging images, are as detailed as possible so that other researchers can better reproduce the method.

Referee #3:

I read the Muzuta et al. manuscript with great interest. The group is probably the best in the field to collect the data need to present this work -focused on the in planta pollen tube growth and ovule targeting process.

I am not critiquing so much on the microscopic imaging aspect, which I think is a monumental project or on the absolute time-resolution or detailed pathways of individual steps. I just want to mention that the authors are probably the only group willing and able to invest the effort - for the community.

When I tried to decipher the imaging details, I did find the details presented on the imaging part overwhelming. However, am not sure how they can be presented in a more accessible way. Interested readers will just have to work through with patience.

I like to focus more on the usefulness of the overall data presented. I believe they are useful to guide the community in resolving events happening inside the pistil. I think the precise minutes will continue to differ somewhat between labs (and these have always been debated with little positive outcome to move our understanding forward), but the trend and relative data presented in some of the data plots are useful references.

e.g. Fig. 6D,7D, are the most systematic collection of data, they will help the community of researchers trying to come up with a consensus mechanism, or multiple mechanisms, for those events monitored here, and in so doing contribute to the synthesis of mechanisms for the not routinely needed event for fertilization success but the elegant "fertilization recovery" event to salvage fertilization.

I think it is an imminently publishable piece of work.

Point-by-point Responses to the Referees' Comments:

Referee #1:

In this article, the authors have used a novel 2D photon live imaging system to directly observe pollen tube behavior. Especially their growth in the transmitting tract, and pollen tube emergence into the ovary. And then how they interact with ovules. 2021). However, since we do not understand the real-time behavior of pollen tube attraction and the temporal relationships between pollen tube and various portions of the pistil, we do not understand how the pollen tubey behaviors are blocked.

Their imaging approach has revealed many new observations - seemingly small to major ones. For example, they have produced the first estimate or enumeration of the number of pollen tubes that grow in a style in vivo. This is small in the larger scheme of things, but the field has operated for a while without accurately knowing this basic information. For major findings, by using a variety of mutants in this assay, they have revealed a repulsion mechanism that is functioning specifically at the pollen tube entry step on the funiculus.

We appreciate the valuable comments and suggestions on our manuscript. All comments kindly provided by referee #1 have been incorporated. Our responses to the specific concerns are highlighted in blue below. The changes have been visually annotated in the main document, accessible via the Track Changes tool in MS Word.

I have two broad feedbacks on this manuscript. The authors should cite major and original findings made by others in the field; I have tried to indicate where possible. Few examples:

- a. When referring to Two-Photon live imaging of pistils, the authors should mention a similar assay being done on the pistils by Rotman et al 2003
- b. Page 14 last sentence: Beale et al 2012 should be cited
- c. Line 4, Page 20: Tsukamoto et al 2010 should also be cited
- d. Line 1, Page 21, Beale et al 2012 should be cited.

The reference to the previous report Rotman et al. (2003) was mentioned in the Introduction section on page 6 line 91 and in the Materials and Methods section on page 30 line 501. We also cited the proposed reports Beale et al. (2012) on page 19 line 325 and page 27 line 452, and Tsukamoto et al. (2010) on page 23 line 380 and page 26 line 439.

Second, the writing is not great in many places and they almost provide no explanation or leave

out crucial details and as a result, the complex findings and issues are not clearly written. I suggest they make a good-faith effort to elaborate on certain sections to articulate their findings even better.

Specific points (arranged by page number and not based on importance). Unfortunately, the journal-provided text file does not have line numbers, so I am using manually counted line numbers within a page and the page number on the document to provide the feedback.

We apologize for the confusing text and inadequate explanations. We also apologize for the inconvenience caused by the lack of line numbers in the text. The entire manuscript has been revised to explain our results with line and page numbers in the revised manuscript.

Line 12, Page 4: 'Arabidopsis thaliana flowers possess single ovaries; is not correct. How can it be single ovaries? Best to describe that the Arabidopsis pistil consists of two fused carpels. Each carpel represents one locule of the ovary that contains many ovules.

We have accepted the reviewer's suggestion and revised this sentence as "In the flower of *Arabidopsis thaliana*, a gynoecium consists of two fused carpels. Each carpel represents one locule of the ovary that contains dozens of linearly aligned ovules." on page 4 line 59 in the revised manuscript.

Line 10, Page 5: regulate is an extra word there and needs to be dropped.

We revised this sentence as "multistep polytubey blocking systems regulate one-to-one pollen tube guidance" on page 5 line 79 in the revised manuscript.

Line 10, Page 5: FERONIA is misspelled with two R; one R needs to be dropped.

This was revised.

Lines 2 and 3, Page 7: Data and conclusions based on that data are being stated. Hence, please reference this conclusion to a figure in the manuscript.

We added the number of ovules per a WT pistil. The number of ovules per a pistil was 46.8 ± 3.3 in the WT *Arabidopsis* (mean \pm s.d.; n = 10 pistils). This information and conclusions based on that data were added on page 7 line 102 in the revised text.

Line 10, Page 7: 'Some pollen tube abruptly separated': How many times this was observed? Otherwise, are they concluding based on the observations of a single pollen tube?

We observed 314 WT pistils using the single-locule method. As a result, pollen tube growth in the TT was observed in 218 of these pistils, and 448 ovules in 110 of these pistils showed pollen tube attraction, which looked like a pollen tube separated from the pollen tube bundle. However, these results are based on observations of about 1/3 of the upper part of the ovary, not the total number observed in the entire ovary. In addition, time-lapse imaging was performed from 1 to 18 HAP, so the number of pollen tube attraction after this time is not included. This information has also been added on page 8 line 129 in the revised manuscript.

Line 10, Page 7: Pollen tube bundle: It would enhance the understanding (especially given the experiments and results in Figure 2), if it is indicated (i) where is the pollen tube emergence site by noting the distance of emergence site from the stigma and (ii) whether this pollen tube was part of the pollen tube bundle or another tube that is behind the pollen tube bundle front that emerged when the pollen tube front has long crossed this pollen tube emergence site.

We defined pollen tube "bundle" as pollen tubes elongating within the transmitting tract on page 7 line 112. We also added yz-projection image as Fig EV1C to mention the attracted pollen tubes. The attracted pollen tubes toward ovules were shown as separated from the pollen tube bundle in the yz-projection images. These explanations were also added in the page 8 line 122.

Line 16, Page 7: Should be rewritten as 'this is the first report of such a dynamic recording of the entire guidance of a single pollen tube.

We accept these suggestions and have edited the text accordingly on page 9 line 136.

Line 8, Page 9: Starting from "Pollen tube behavior and growth path". From this point onwards, it would help to have this be part of a separate paragraph. Otherwise, it is a really long paragraph with multiple results/outcomes being discussed.

We accept these suggestions and have edited the text accordingly on page 11 line 170.

Line 11, Page 9: About the pollen tube interaction with SE: How many times this was observed? Otherwise, are they concluding based on the observations of a single pollen tube?

We observed similar pollen tube movements within the transmitting tract in the many pistils examined. Such movements were observed even in the presence of multiple pollen tubes. As examples, we mentioned on page 11 line 176 in the revised text as white arrow in Fig 3D, and cyan, green, and orange arrows in Fig 4B.

Line 4, Page 10: Figure 4 A reference: What is the number of emerged pollen tubes from which this conclusion is based? Is it 11 tubes, as mentioned below?

Two-photon live imaging of the 110 WT pistils for 1 to 18 HAP revealed 448 ovules with pollen tube attraction. Of these, 11 pollen tubes in 7 pistils were able to measure the growth rate of pollen tube within the TT at least 2.5 hours before pollen tube emergence. This sentence was added on the page 12 line 193 in the revised text.

Page 10: After the authors mention the results on pollen tubes that bind SE (Figure 4B-C and Figure S3B) and have not said anything, I am left wondering what happens to the growth rate of the tubes in wild-type pistils after they emerge. Why not describe it here so that there is a comprehensive and complete understanding of what happens to the tubes in the wild type? Instead of breaking this apart like what was done here, and the remainder be shown later after describing the results in Figure 5 (first few sentences at the top of page 14)?

The growth rates of the 9 emerged pollen tubes (as shown in Fig 4E) on both septum and funiculus after pollen tube emergence were measured from yz-projection images. These data were added on page 13 line 204 in the revised text. The summarized sentence was also added at the top of next section "Long-distance guidance signal depending..." in page 14 line 231.

Page 12: In this whole paragraph, for each genotype, it might be better to mention what they have and what they lack, as the goal is to discern the individual roles of sporophytic and gametophytic tissues. For example,

1. WT has both sporophytic and gametophytic tissues in an ovule.
2. In *ant*, both sporophyte and gametophyte are severely affected.
3. In *dif1*, they lack gametophyte even though the sporophytic tissues are present.
4. In the *ino* mutant, they lack outer integument of the ovule.

Stating this way helps when concluding that since pollen tube emergence was observed in 1 and 3, sporophytic tissues play a key role in pollen tube emergence from the TT. Without a clear

description as to what each genotype lacks and contains, it is not clear why the conclusion was drawn based on the observations.

We accept these suggestions and have revised each mutant phenotypes as (1) WT, (2) *ant*, (3) *dif1*, and (4) *ino* in page 15 line 241. The results were also mentioned by these numbers, and the figures for the mutants in Fig 5 have been rearranged so that they appear in the order in the revised text. To explain the results in more detail, the tables were changed to graphs as Fig 5B, C and D with statistical analysis in the revised manuscript.

The penultimate sentence (2nd from the bottom) on Page 13: Stating that 'pollen tube emergence in the *ino* ovary was assessed through two-photon live imaging" confused me. Up until this point in this section, I assumed that the results shown in Figure 5 were all based on two-photon live imaging. However, the authors mention here that they used two-photon live imaging only to confirm the results in Figure 5. Upon reading the figure legend, it seems the observations were made with cleared pistils. Therefore, the authors should make a special mention of what technique was used when describing the results shown in Figures 5 and 4.

We mentioned that ClearSee-treated cleared pistils were used on page 15 line 240. We also noted in the revised text whether the method used in each experiment was live imaging or cleared fixed pistils.

Line 3, Page 16: NaOH-cleared pistils: Make a distinction like this throughout the manuscript. In the text when describing the experiments involving each mutant, they should mention, whether the analysis was using cleared pistils or two-photon live imaging.

We revised it that way throughout the manuscript.

The fourth line from the bottom of Page 16: Since FER at 4 HAP (but not LRE mutant) showed that more tubes are emerging from the TT into the ovary, how about stating that FER mainly controls the polytubey at the time of pollen tube emergence in the septum (like the way they said they are testing this hypothesis at the beginning of this paragraph)?

We accept these suggestions and have edited the text accordingly on page 21 line 352.

Page 16, section title (3 sentences from the bottom): The section title sentence is incomplete. More words following the words 'pollen tube enters' must be added:

"...the ovary or first pollen tube emergence"

We accept these suggestions and have added the section title as "...pollen tube enters the funiculus" on page 22 line 360.

Line 5, Page 17: How do these rates compare to the *in vivo* rates of polytubey in the first reports made on these mutants (Huck et al 2003, Tsukamoto et al 2010)? The authors should cite those numbers to both show how well their observations compare with those and also to cite those references for making these observations *in vivo*.

We compared *in vivo* rates of polytubey in the *fer* and *lre* mutants by citing three references (Huck et al., 2003; Tsukamoto et al., 2010; Zhong et al., 2022) on page 22 line 373 in the revised text.

Page 18: "The extent of the polytubey block became stricter 45 min after the entry of the first pollen tube"

Could there be an alternate explanation for pollen tubey block becoming stricter? Since there are only ~70 tubes (their own data at the beginning of the manuscript and there are not enough tubes to get more than two tubes per ovule and therefore it appears as if the block is intensifying after the entry of the first pollen tube. Hence this sentence should be changed to simply reflect the observation rather than imply that the block became stricter.

We accept these suggestions and have edited the text accordingly on page 24 line 399. We have also revised the text to make it clearer that the first block is defective, and the second block is incomplete but present in both *fer* and *lre* mutants.

Figure 4C. The word 'observation' is misspelled on the X-axis of the graph

This was revised.

Referee #2:

The manuscript entitled 'Deep imaging reveals dynamics and signaling in one-to-one pollen tube guidance' by Yoko Mizuta et al. reported the study on the functional mechanisms underlying the directional cues and polytubey blocks in the depths of the pistil. The authors developed a "single-locule method" using two-photon excitation microscopy to perform deep

imaging of pollen tube guidance in living ovaries of *A. thaliana*. Using novel imaging methods, the authors found that the emergence of pollen tubes is stochastically determined by the distribution of pollen tubes. Mutant studies involving *dif*, *ant* and *ino* revealed that ovular outer integument-dependent signals may enhance pollen tube emergence and ovule maturation contributes to ovule targeting. Further results demonstrated that FER-dependent sporophytic signaling as a polytubey block by restricting the number of emerged pollen tubes onto the septum. Additionally, gametophytic FER- and LRE-dependent signaling regulates polytubey block on the funiculus, even before the arrival of the first pollen tube at the synergid cell that exerts the blocking signal. The author also revealed FERONIA- and LORELEI-independent repulsion signal involved in polyubey blocks on the ovular funiculus. The authors conclude that multiple signals from the ovules are involved in achieving one-to-one guidance, with polytubule blocks occurring through multiple steps. The methodology established by the authors for the investigation of in-vivo pollen tube guidance holds significant implications for advancing research in this area. Therefore, it is of general interest and worthy of publication in EMBO Reports.

We appreciate the valuable comments and suggestions on our manuscript. All comments kindly given by referee #2 were incorporated. Our responses to the specific concerns are highlighted in blue below. The changes have been visually annotated in the main document, accessible via the Track Changes tool in MS Word.

Comments:

1. In Figure 4E, the emerged pollen tubes can grow on the funiculus in *ino* pistils, but there was a significantly reduced number of ovules showing funicular guidance in Figures 5A and 5B for *ino* mutant ovaries. It is necessary for the authors to discuss the possible reasons for this geographical difference.

It is easy to distinguish between pollen tube elongation at the septum and the funiculus in the WT, whereas it cannot be defined in the *ino* pistil. Therefore, we accept these suggestions and have revised Figure 4E to add a schematic diagram of the *ino* pistil and changed the green line to a gray line, which indicates the pollen tube elongation in the locule without attachment to the female tissues.

2. In Page 12, the author mentioned that "In *ant* mutant ovaries, there was a reduced number of emerged pollen tubes and ovules showing funicular guidance" this conclusion is not well reflected in Figure 5B. I advise that the authors confirm whether the statement is correct.

This sentence was revised as "In *ant* mutant ovaries, there was a similar number of emerged pollen tubes but significantly reduced number of ovules showing funicular guidance" on page 16 line 263 in the revised manuscript. Statistical analysis was also performed to explain this result.

3. The experimental steps of two-photon imaging, especially the post-processing of the imaging images, are as detailed as possible so that other researchers can better reproduce the method.

We are pleased to learn that referee #2 finds our two-photon method useful for other researchers. We hope that this will be a method to accelerate the study of plant reproduction.

Referee #3:

I read the Muzuta et al. manuscript with great interest. The group is probably the best in the field to collect the data need to present this work -focused on the in planta pollen tube growth and ovule targeting process.

I am not critiquing so much on the microscopic imaging aspect, which I think is a monumental project or on the absolute time-resolution or detailed pathways of individual steps. I just want to mention that the authors are probably the only group willing and able to invest the effort - for the community.

When I tried to decipher the imaging details, I did find the details presented on the imaging part overwhelming. However, am not sure how they can be presented in a more accessible way. Interested readers will just have to work through with patience.

I like to focus more on the usefulness of the overall data presented. I believe they are useful to guide the community in resolving events happening inside the pistil. I think the precise minutes will continue to differ somewhat between labs (and these have always been debated with little positive outcome to move our understanding forward), but the trend and relative data presented in some of the data plots are useful references.

e.g. Fig. 6D,7D, are the most systematic collection of data, they will help the community of researchers trying to come up with a consensus mechanism, or multiple mechanisms, for those events monitored here, and in so doing contribute to the synthesis of mechanisms for the not routinely needed event for fertilization success but the elegant "fertilization recovery" event to

salvage fertilization.

I think it is an imminently publishable piece of work.

We are pleased to learn that referee #3 finds our study interesting and the importance of this work. We apologize for the confusing text and insufficient explanations to understand the results. We have reviewed the entire manuscript and added the necessary descriptions to explain our results in the revised manuscript. We hope that this manuscript will accelerate the study of plant reproduction.

Dear Dr. Mizuta

Thank you for the submission of your revised manuscript to EMBO reports and for addressing all referee concerns in the manuscript and point-by-point response.

Browsing through the manuscript, I noticed a few editorial things that we need before we can proceed with the official acceptance of your study.

- Please describe your findings in the Abstract in present tense.
- You might want to change the statement in the Data Availability section to "This study includes no data deposited in external repositories". Have you considered depositing your timelapse imaging data to a public repository such as Image Data Resource or Biolineage Archive? You could deposit the raw, full imaging data you have instead of 'only' the z-projections, so that other scientists can make full re-use of it.
- Please update the 'Conflict of interest' paragraph to our new 'Disclosure and competing interests statement'. For more information see <https://www.embopress.org/page/journal/14693178/authorguide#conflictsofinterest>
- Regarding the Author Contributions, we now use CRediT to specify the contributions of each author in the journal submission system. Therefore, please remove the Author Contributions from the manuscript file and make sure that the author contributions in our manuscript tracking system are correct and up-to-date. The information you specified in the system will be automatically retrieved and typeset into the article. You can enter additional information in the free text box provided, if you wish.
- Callouts to Fig 1D, E are missing in the text. Fig 2G-J are called out in the movie legend, but Fig 2 only has A-F. Please check.
- Please remove the legend of Table EV1 from the manuscript file - it is already provided in the file itself.
- Please remove the movie legends from the manuscript file as they are already provided correctly in each zip file.
- Thank you for the submission of Cover suggestions. The images look beautiful indeed. We will consider your suggestions when we decide on the cover of the issue your study will be published in.
- Our production/data editors have asked you to clarify several points in the figure legends (see below). Please incorporate these changes in the manuscript and return the revised file with tracked changes with your final manuscript submission.
 - a) Please note that a separate 'Data Information' section is required in the legends of figures 1b-g; 4b, d; 6a-c; EV 1a-f; EV 2b-c.
 - b) Please note that the scale bar information for figure EV 4a is incorrectly labelled as EV 4b. This needs to be rectified.
 - c) Please note that the figure EV 4d is missing in the manuscript, however the legend for the same is provided in the manuscript. This needs to be rectified.
 - d) Please note that information related to n is missing in the legend of figure 6c.
 - e) Please note that the scale bar needs to be defined for figures EV 2a-c.
 - f) Please note that the arrow and arrowheads are not defined in the legend of figure EV 1b, d. This needs to be rectified.
 - g) Please note that the arrows are not defined in the legend of figure EV 4a. This needs to be rectified.
- Finally, EMBO Reports papers are accompanied online by A) a short (1-2 sentences) summary of the findings and their significance, B) 2-3 bullet points highlighting key results and C) a synopsis image that is 550x300-600 pixels large (width x height) in PNG or JPG format. You can either show a model or key data in the synopsis image. Please note that the size is rather small and that text needs to be readable at the final size. Please send us this information along with the revised manuscript.

- On a different note, I would like to alert you that EMBO Press offers a new format for a video-synopsis of work published with us, which essentially is a short, author-generated film explaining the core findings in hand drawings, and, as we believe, can be very useful to increase visibility of the work. This has proven to offer a nice opportunity for exposure i.p. for the first author(s) of the study. Please see the following link for representative examples and their integration into the article web page:
https://www.embopress.org/video_synopses
<https://www.embopress.org/doi/full/10.15252/emboj.2019103932>

With kind regards,

All editorial and formatting issues were resolved by the authors.

Dr. Yoko Mizuta
Nagoya University
Institute of Transformative Bio-Molecules (WPI-ITbM)
Furo-cho
Chikusa-ku
Nagoya, Aichi 464-8601
Japan

Dear Yoko,

I am very pleased to accept your manuscript for publication in the next available issue of EMBO reports. Thank you for your contribution to our journal.

Kind regards,

Martina
